



# Organic carbon burial by river meandering partially offsets bank-erosion carbon fluxes in a discontinuous permafrost floodplain

Madison M. Douglas[1], Gen K. Li[1], Woodward W. Fischer[1], Joel C. Rowland[2], Preston C. Kemeny[1], A. Joshua West[3], Jon Schwenk[2], Anastasia P. Piliouras[2], Austin J. Chadwick[1], Michael P. Lamb[1]

[1]Division of Geological and Planetary Science, California Institute of Technology, Pasadena, CA, 91125, USA
[2]Earth and Environmental Sciences Division, Los Alamos National Laboratory, Los Alamos, NM, 87545, USA
[3]Department of Earth Sciences, University of Southern California, Los Angeles, CA, 90089, USA

*Correspondence to*: Madison M. Douglas (mmdougla@caltech.edu)

**Abstract.** Arctic river systems erode permafrost in their banks and mobilize particulate organic carbon (OC). Meandering rivers can entrain particulate OC from permafrost many meters below the depth of annual thaw, potentially enabling OC oxidation and the production of greenhouse gases. However, the amount and fate of permafrost OC that is mobilized by river erosion is uncertain. To constrain OC fluxes due to riverbank erosion and deposition, we collected riverbank and floodplain sediment samples along the Koyukuk River, which meanders through discontinuous permafrost in central Alaska. We measured sediment total OC (TOC), radiocarbon content, water content, bulk density, grain size, and floodplain stratigraphy. Radiocarbon abundance and TOC were higher in samples dominated by silt as compared to sand, which we used to map OC content onto floodplain stratigraphy and estimate carbon fluxes due to river meandering. Results showed that sediment being eroded from cutbanks and deposited as point bars had similar OC stocks (mean±1SD of 125.3±13.1 kgOC m$^{-2}$ in cutbanks versus 114.0±15.7 kgOC m$^{-2}$ in point bars) whether or not the banks contained permafrost. We also observed radiocarbon-depleted biospheric OC in both cutbanks and permafrost-free point bars. These results indicate that a significant fraction of aged biospheric OC that is liberated from floodplains by bank erosion is subsequently re-deposited in point bars, rather than being oxidized. The process of aging, erosion, and re-deposition of floodplain organic material may be intrinsic to river-floodplain dynamics, regardless of permafrost content.

## 1 Introduction

The warming climate is changing Arctic landscapes, inducing complex feedbacks in the global carbon cycle as permafrost soils thaw (Schuur et al., 2015; Turetsky et al., 2020). Changes in air temperature and precipitation have increased the thickness of the active layer (ground overlying permafrost that experiences seasonal freeze-thaw cycles), allowing respiration of soil organic carbon (OC) previously frozen for thousands of years (Romanovsky et al., 2010; Isaksen et al., 2016; Biskaborn et al., 2019). Organic carbon is also lost from permafrost through erosion by Arctic rivers—the six largest Arctic rivers contribute ~3 Tg of river particulate OC (POC) to the Arctic Ocean annually (McClelland et al., 2016). Since a substantial portion of



eroded POC is thought to be prone to oxidation (Schreiner et al., 2014), river erosion of POC could play an important role in the greenhouse gas fluxes associated with permafrost thaw (Toohey et al., 2016; Walvoord and Kurylyk, 2016).

As Arctic rivers migrate laterally across permafrost floodplains containing high concentrations of soil OC, they mine sediment and organics from tens of meters below the active layer (Spencer et al., 2015; Kanevskiy et al., 2016). Permafrost banks are

thus an important source of POC to rivers (Kanevskiy et al., 2016; Loiko et al., 2017; Lininger et al., 2018; Lininger and Wohl, 2019). After mobilization by a river, POC can be oxidized during transport (Striegl et al., 2012; Denfeld et al., 2013; Serikova et al., 2018) or re-buried in floodplains (Wang et al., 2019; Torres et al., 2020). Alternatively, POC can be delivered downstream to the ocean, where it may be oxidized to $CO_2$ or $CH_4$ or buried in deltaic sedimentary deposits (Torres et al., 2020; Hilton et al., 2015). Riverbank erosion may be limited by the rate of permafrost thaw (Costard et al., 2003;

Randriamazaoro et al., 2007; Dupeyrat et al., 2011), implying that erosion rates could increase with warming air and river water temperatures. Therefore, more rapid riverbank erosion has the potential to generate a significant climatic feedback by making POC previously frozen in permafrost available for oxidation (Striegl et al., 2012; Denfeld et al., 2013; Serikova et al., 2018), but the magnitude of this feedback is highly uncertain.

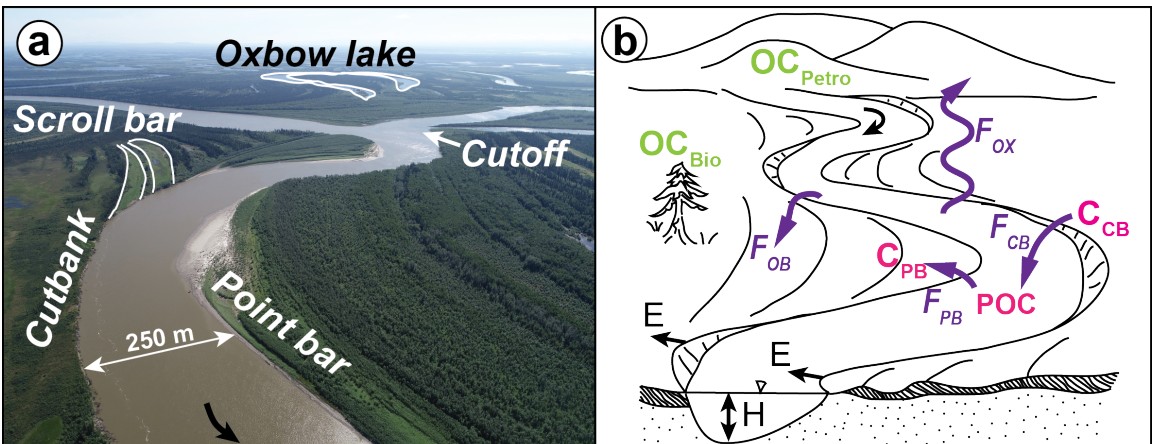


**Figure 1: Overview of sediment erosion and deposition patterns in meandering river floodplains and important variables influencing the regional carbon cycle. (a) Drone photograph taken overlooking the Koyukuk River floodplain, Alaska. The river flows south toward the bottom of the image (indicated by black arrow), eroding the cutbank on the outside of the river bend and depositing sediment on the point bar. Channel migration generates bands of higher and lower elevation sections of floodplain called scroll bars.**

**As the river migrates, an individual bend becomes more sinuous, eventually cutting itself off and abandoning a section of channel, which becomes an oxbow lake. (b) Schematic of a meandering river floodplain, with channel geometry variables shown in black and particulate organic carbon reservoirs and fluxes into and out of the river control volume shown in purple. The river has bankfull depth $H$ and migrates laterally at rate $E$, maintaining a constant channel width. Organic carbon is stored in the river cutbanks ($C_{CB}$) and point bars ($C_{PB}$), and is transported in the river as particulates (POC). These reservoirs are mixtures of radiocarbon-dead (Fm**

**= 0) petrogenic organic carbon ($OC_{Petro}$) and biospheric organic carbon ($OC_{Bio}$) that has been stored in permafrost (low Fm) or been recently fixed by the biosphere (Fm ≥ 1). Fluxes of organic carbon into and out of the river control volume include cutbank erosion ($F_{CB}$), point bar deposition ($F_{PB}$), overbank deposition ($F_{OB}$), and oxidation of POC and DOC ($F_{OX}$).**





Recent work has studied floodplain POC stocks vulnerable to erosion by Arctic rivers (Vonk et al., 2019; Parmentier et al., 2017). For instance, Lininger et al. conducted an extensive field campaign to map OC concentrations and stocks across the

Yukon Flats, and found statistically significant variability in OC concentrations between geomorphic landforms produced by river processes (Lininger et al., 2018) as well as systematic underestimation of floodplain OC stocks in large data compilations (Lininger et al., 2019). Their work built on previous studies that characterized vegetation and permafrost succession through a time series of floodplain surfaces that had been progressively abandoned by river migration (Shur and Jorgenson, 2007). Yet major questions remain about the magnitude of POC fluxes due to bank erosion and bar deposition in permafrost river systems,

as well as the physical processes that govern these fluxes (Lininger and Wohl, 2019). Alluvial rivers commonly maintain an approximately constant channel width, eroding one bank while depositing sediment at a commensurate rate on the opposite bank (Fig. 1a) (Dietrich et al., 1979; Eke et al., 2014). Previous work on meandering rivers demonstrates that rapidly eroding permafrost bluffs may contribute significantly to downstream POC fluxes (Kanevskiy et al., 2016). However, it is unclear to what extent the OC released by bank erosion is compensated by OC burial in depositional bars, as opposed to being transported

downstream or oxidized during transport within river systems (Fig. 1b) (Wang et al., 2019; Scheingross et al., 2021).

To quantify POC storage and mobilization in Arctic floodplains, we investigated the Koyukuk River in Alaska (Fig. 2), which is an actively meandering river in discontinuous permafrost. We quantified OC stocks using field observations of permafrost occurrence and floodplain stratigraphy to extrapolate laboratory measurements of sediment grain size and total OC. We then

used a one-dimensional mass-balance model to quantify net fluxes of OC into the river due to bank erosion and bar deposition. To attribute OC to biospheric versus rock-derived (petrogenic) sources, we used radiocarbon measurements to infer the abundance of a petrogenic OC end-member and calculate the radiocarbon fraction modern of biospheric carbon in permafrost and non-permafrost sediment.

## 2 Measurements and approach

To understand cycling of POC between rivers and floodplains, we developed an approach to ascertain OC sources and determine if OC eroded from river deposits is transported downstream or reburied (Fig. 1b). Eroding banks can source OC from modern vegetation and organic horizons near the bank surface as well as deeper sediment that may be depleted in radiocarbon. Radiocarbon provides an effective tracer of OC aging in floodplains (Galy and Eglinton, 2011; Torres et al., 2017), but several processes can produce depleted radiocarbon signals. First, Arctic permafrost deposits are mostly relict, with

low fractions of modern radiocarbon (Fm = $[^{14}C/^{12}C]_{sample}/[^{14}C/^{12}C]_{modern}$) (O'Donnell et al., 2012). If mobilized permafrost POC is re-buried in bars without the addition of newly fixed biospheric OC, then bar sediment should also have OC with low Fm inherited from permafrost carbon. Second, sediment can contain a radiocarbon-dead, petrogenic OC component that contributes to low Fm values (Blair et al., 2003). We expected a petrogenic OC contribution in floodplain sediments throughout the Koyukuk River system, since the headwaters of the Koyukuk River contain outcrops of shale bedrock rich in kerogen that

source oil to the Prudhoe Bay oilfields (Dumoulin et al., 2004; Wilson et al., 2015; Slack et al., 2015). Third, river-floodplain



interactions generate low Fm carbon via transient OC storage, independent of the presence of either permafrost or petrogenic OC (Torres et al., 2020). For example, floodplain deposits can remain in place over millennial timescales before being reworked by the river channel due to the stochastic nature of river lateral migration (Torres et al., 2017; Repasch et al., 2020). Therefore, radiocarbon measurements provide insight into OC sources, but require de-convolving petrogenic OC from

biospheric OC, and assessing aging of OC by storage in permafrost versus non-permafrost floodplain deposits.

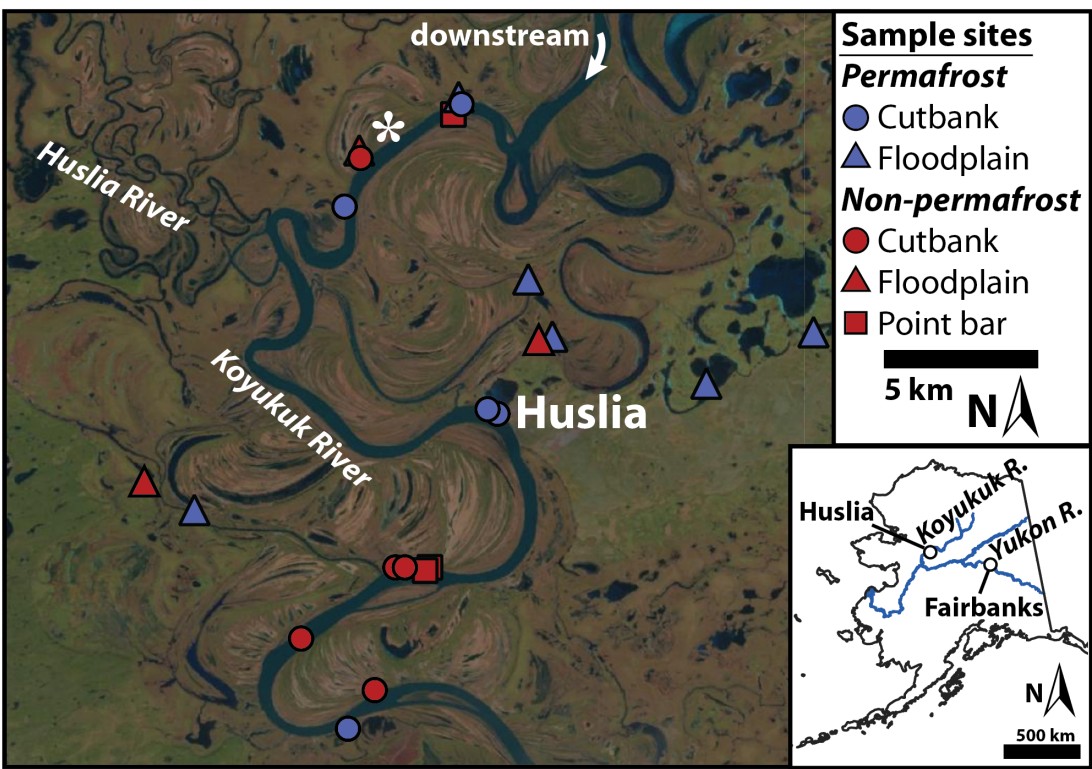

**Figure 2: Sample locations on the Koyukuk River floodplain. Locations are coded for sites where we sampled ice-cemented permafrost versus ice-poor ground inferred to be non-permafrost. Sample sites are located near the village of Huslia, in central**
**Alaska, and the river flows towards the south past town. Sampling locations are mapped on Landsat imagery, with the white star marking the location of Fig. 1a (drone photo taken looking east). The inset map was generated using the "Alaska Coast Simplified" and "Major Rivers" shapefiles from the Alaska State Geo-Spatial Data Clearinghouse.**

We used sediment TOC and Fm measurements to calculate the Fm of biospheric and contribution from petrogenic OC end-
members (Sect. 2). This calculation allowed us to determine if low Fm values were due to a high concentration of radiocarbon-dead rock-derived OC, or preservation and aging of OC in permafrost or in the river floodplain (Fig 1b) (Scheingross et al.,





2021). Both radiocarbon-dead OC derived from bedrock erosion ($TOC_{petro}$) and aging of biosphere-derived OC ($TOC_{bio}$) in permafrost and river floodplain deposits will yield sediment OC with low Fm (Fig 1b). We partitioned the TOC measured in each sample ($TOC_{meas}$) into a two end-member mixture of biospheric ($f_{bio}$) and petrogenic OC ($f_{petro}$) fractions, with a constant mass fraction of petrogenic OC (Fig. 4d) (Blair et al., 2003; Cui et al., 2016):

$$1 = f_{bio} + f_{petro}, \tag{1}$$

$$TOC_{meas} = f_{bio}TOC_{meas} + f_{petro}TOC_{meas}. \tag{2}$$

Changes in the ratio of biospheric to petrogenic OC, as well as aging of the biospheric pool, will change the measured fraction modern in sediment OC ($Fm_{meas}$; unitless ratio) (Galy et al., 2008). By mass balance,

$$Fm_{meas} = f_{bio}Fm_{bio} + f_{petro}Fm_{petro}, \tag{3}$$

The petrogenic OC end-member was assumed to be radiocarbon-dead ($Fm_{petro} = 0$), and Eqs. (1) and (2) substituted into Eq. (3):

$$Fm_{meas} = \frac{Fm_{bio}(TOC_{meas} - f_{petro}TOC_{meas})}{TOC_{meas}}. \tag{4}$$

A regression of Eq. (4) for $Fm_{meas}$ versus $TOC_{meas}$ was used to calculate the $Fm_{bio}$ (effectively the mean age of biosphere-derived carbon) and the mass fraction $f_{petro}$ (Hemingway et al., 2018; Wang et al., 2019). We assumed that petrogenic OC concentration in floodplain sediment is constant along our study reach ($TOC_{petro} = f_{petro} \times TOC_{meas}$ is constant for all stratigraphic units). While recent work found evidence for petrogenic OC oxidation during riverine transport of sediment (Bouchez et al., 2010; Horan et al., 2019), these studies focused on river reaches spanning hundreds of kilometres, an order of magnitude longer than our study reach. Even over hundreds of kilometers, Horan et al. (2019) found that less than half of petrogenic OC eroded from the Mackenzie River catchment was oxidized during transport. Therefore, it is reasonable to assume that the production and oxidation of significant rock-derived OC is minor within our study reach.

To separate biospheric OC that was produced *in situ* versus eroded from a cutbank, transported as POC and re-deposited by the river, we use a linear mixing model following Scheingross et al. (2021):

$$f_{bio,is} = \frac{Fm_{bio} - Fm_{bio,cb}}{Fm_{bio,is} - Fm_{bio,cb}}, \tag{5}$$

Where $f_{bio,is}$ is the fraction of biospheric OC produced in situ, $Fm_{bio}$ is the fraction modern of biospheric OC for each sediment sample, $Fm_{bio,cb}$ is the cutbank OC end-member, and $Fm_{bio,is}$ is the *in situ* biospheric OC endmember (Supplemental Table S6). We select a modern topsoil with the highest measured $Fm_{bio}$ as the *in situ* OC endmember ($Fm_{bio,is}$; KY18-Core5-15), and the oldest woody debris from a cutbank as the cutbank endmember ($Fm_{bio,cb}$; KY18-Bank14).



## 3 Materials and methods

### 3.1 Field sampling methods

We studied deposits and collected samples from 33 locations along the Koyukuk River near the village of Huslia, Alaska, during June – July 2018 (Fig. 2 inset; Supplemental Fig. S1). Near Huslia, the mean annual air temperature is -3.6 °C (Nowacki et al., 2003; Daly et al., 2015, 2018). The Koyukuk is a meandering river in discontinuous permafrost with well-defined scroll bars (former levees) (Mason and Mohrig, 2019) that demarcate clear spatial patterns of channel lateral migration (Fig. 2) (Shur and Jorgenson, 2007). Seasonal variations in temperature cause an annual freeze-thaw cycle in sediment near the ground surface across the landscape, called the active layer, while the ground below consists of permafrost or is perennially unfrozen. To represent the diversity of floodplain geomorphology, permafrost occurrence, and deposit ages, we selected 8 permafrost cutbanks, 6 non-permafrost cutbanks, 6 permafrost floodplain cores, 4 non-permafrost floodplain cores and pits, and 9 non-permafrost cores and pits in transects across 2 point bar complexes to characterize floodplain stratigraphy and carbon geochemistry (Fig. 2; Supplemental Tables S1 & S2). We categorized permafrost as ice-cemented sediment observed during our summer field season, often containing ice lenses and other structures indicative of permafrost (Fig. 3a-b) (French and Shur, 2010). Permafrost cutbanks often had an undercut marking the high water level where bank sediment was directly thawed by the river and collapsed as well as abundant toppled trees indicating active bank erosion. We classified terrain without ice cement observed to the depth of coring or sampling as non-permafrost (Fig. 3a, c). Bands of vegetation outlined scroll bars on the floodplain that were abandoned due to channel lateral migration and meander-bend cutoff. Mean bank erosion rates for the portion of the Koyukuk we studied were 0.52 m yr$^{-1}$ from 1978-2018 (Rowland et al., 2019). Over the same time interval, channel width varied from 173±43 m in 1978 to 179±43 m in 2018 (median±1SD), indicating a balance between cutbank erosion and point bar deposition over this period since net erosion or deposition would change channel width (Supplemental Fig. S2).

River bathymetry was characterized using a Teledyne RioPro acoustic Doppler current profiler (ADCP). We calculated a river depth of 12.4 as the mean of the deepest measured value for 8 ADCP river cross-sectional transects across a representative meander bend. Bank samples were collected by digging into cutbanks and point bars, and cores were taken using a hand auger in non-permafrost deposits and a Snow, Ice, and Permafrost Research Establishment (SIPRE) corer in permafrost (Fig. 2). All samples were recorded in stratigraphic columns to determine the thickness of each stratigraphic unit. Samples were stored in sterile Whirlpak bags and frozen within 12 hours of collection, then transported frozen back to a cold room (-15°C) at Caltech for laboratory analyses.

### 3.2 Laboratory analyses

Samples were transferred to pre-combusted aluminium foil, weighed on a laboratory scale, and oven dried at 55-60°C to calculate the mass fraction of water ($M_{H2O,i}$). For samples taken using the SIPRE core with known volume, bulk density ($\rho_i$)



was calculated from total mass divided by volume. The samples were gently homogenized using an agate mortar and pestle, then split using cone-and-quarter or a riffle splitter for further analysis.

Total organic carbon (TOC, $TOC_{meas}$ in Eq. 2) and total nitrogen (TN) were measured on a Costech Elemental Analyzer coupled to a MAT 253 IRMS at Los Alamos National Laboratory (LANL). Prior to analysis, approximately 3 mg of each sample was decarbonated by fumigation with HCl in silver capsules. Isotope ratios are reported relative to the Vienna Pee Dee Belemnite (VPDB; $\delta^{13}C = (R_{sample}/R_{VPDB}-1) \times 1000$; reported in per mille (‰)) and measured blanks were below peak detection limit. Measurements were calibrated using laboratory standards of 25-(Bis(5-tert-butyl-2-benzo-oxazol-2-yl) thiophene (BBOT,

Eurovector; TOC = 72.53%, measured as 69.59±2.05%; $\delta^{13}C$ = -26.6‰, measured as -26.6±0.01‰; TN = 6.51%, measured as 6.82±0.24%), Peach Leaves (1570a; TOC = 44.65%; measured as 44.33±0.96%; $\delta^{13}C$ = -25.95‰, measured as -26.13±0.08‰; TN = 2.83%, measured as 3.31±1.27%), and Urea (Eurovector; TOC = 20.00%, measured as 17.98±0.37%; TN = 46.65%, measured as 45.88±0.88%) for TOC and TN, and cellulose (IAEA-C3; $\delta^{13}C$ = -24.91‰, measured as -24.82±0.06‰), sucrose (IAEA-C6; $\delta^{13}C$ = -10.8‰, measured as -10.7±0.03‰) and oxalic acid (IAEA-C8; $\delta^{13}C$ = -18.3‰,

measured as -18.5±0.06‰) for stable C isotopes, with uncertainties reported as 1 standard deviation (±1SD). Stable OC isotope and TN measurements are not discussed in the main text but are included in supplemental figures and tables.

Sample splits for radiocarbon were decarbonated at Caltech in pre-combusted glassware using 1M HCl, sonicated for 10 min, and neutralized using 1M NaOH. Splits were then centrifuged for 10 min, and had the supernatant removed using a pipette.

The samples were then rinsed using 20 mL Milli-Q water, centrifuged and decanted twice before being lyophilized and sent to the National Ocean Sciences Accelerator Mass Spectrometry (NOSAMS) facility in Woods Hole for radiocarbon ($Fm_{meas}$ in Eq. 3), total organic carbon (dry wt% with 5% measurement uncertainty) and organic carbon stable isotope measurements (referenced to VPDB; $\delta^{13}C = (R_{sample}/R_{VPDB}-1) \times 1000$; reported in per mille (‰)). A comparison of OC measurements between NOSAMS and LANL is plotted in Supplemental Figure S5.


Samples for grain size analysis were split using a riffle splitter and placed into sterile polypropylene Falcon tubes to remove carbonate and organic materials (Gee and Or, 2002). Samples were acidified overnight with 1M HCl, then centrifuged for 15 min at 4,000 rpm and decanted; rinsed twice with DI $H_2O$, centrifuged and decanted before being oven-dried at 55-60°C; and then reacted with $H_2O_2$ on a hot plate at 85°C to remove organics. Floating pieces of organic material were removed using a

microspatula rinsed with DI $H_2O$. Additional $H_2O_2$ was added until reactions ceased by visual inspection. Samples were rinsed and centrifuged three times before oven drying. Each sample was re-hydrated using DI $H_2O$, Calgon was added to prevent flocculation, and samples were sonicated for 3 min. The samples were split while wet and grain size was measured using laser diffraction on a Malvern Mastersizer 2000, with measurements calibrated against a laboratory silica carbide standard ($D_{50}$ = 13.184 ± 0.105 μm throughout our measurements). Grain size data were used to validate field observations of grain size that

were made using a sand card (Supplemental Table S5).

Earth **Surface**
**Dynamics**
Discussions

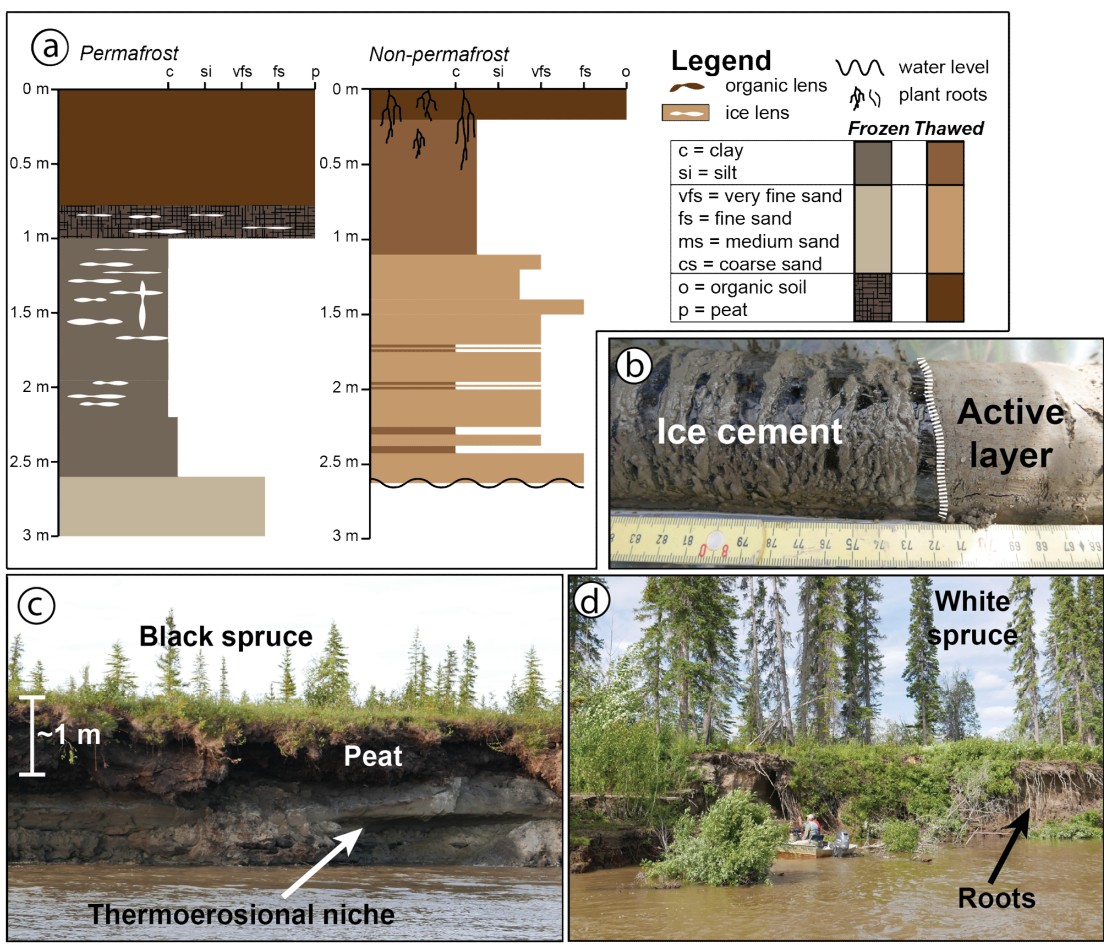

**Figure 3: Field observations of Koyukuk floodplain sedimentology. (a) Representative stratigraphic columns from non-permafrost (Bank 2) and permafrost (Bank 6) cutbanks. (b) Field photo of boundary between permafrost ice cement and the overlying active layer in Core 4. (c) Thermoerosional niche formed in a permafrost cutbank, with silty permafrost overlain by a layer of peat and black spruce trees. (d) Eroding riverbank without permafrost, hosting a white spruce forest with roots that reach deep into the bank sediment. Complete stratigraphic sections and additional field photos are in Supplemental Figures S2-3.**

## 4 Results

Permafrost cutbanks and floodplains generally displayed an organic-rich upper horizon, which extended up to 1.3 m below the ground surface in peat, underlain by silt that abruptly transitioned to sand (Fig. 3a, d; Supplemental Fig. S3). The thickness of the active layer, measured by trenching or using a 1 m permafrost probe (n=53), ranged from 40 cm to greater than the length



of the probe, with a median of measured values (n=38) of approximately 75 cm. Non-permafrost cutbanks had a layer of organic topsoil overlying silt with abundant roots and organic-rich lenses that became interbedded and then transitioned to sand with increasing depth (Fig. 3a). All terrain types exhibited a trend of grain size fining upward, with medium sand (based on bed-material grab samples taken from a boat with a Ponar sampler) comprising the channel bed. We did not observe

permafrost in active point bars, which had a thin to absent layer of organic topsoil at the land surface underlain by sandy deposits exhibiting ripple and dune cross stratification that indicated active sediment re-working and deposition. Sediment TOC and radiocarbon Fm measurements varied with sediment size. Silt samples had higher average TOC than sandy samples, and peat had higher TOC than topsoil (Fig. 4a). Although the organic horizons overlying permafrost had a higher TOC, sediment samples below the organic horizon did not show a significant difference in TOC based on the presence or absence of

permafrost for a given grain size (Fig. 4a-b). The strong dependence of TOC on grain size allowed us to calculate OC stocks based on measured stratigraphic sections.

Coarser sediment yielded lower radiocarbon Fm—indicative of older organic carbon—with silt and organic horizons having higher Fm values (Fig. 4c). A petrogenic contribution can explain measured differences in sediment Fm and would be expected

to be enriched in the coarser size fraction (Galy et al., 2007). To calculate the petrogenic and biospheric end-members for soil OC, we fit the relationship between $Fm_{meas}$ and $TOC_{meas}$ using Eq. (4) and the Matlab 2017 function nlinfit.m, using iterated fitting to calculate 95% confidence intervals. Fitting $Fm_{meas}$ to $TOC_{meas}$ gave biospheric radiocarbon ($Fm_{bio}$) and petrogenic OC content ($TOC_{petro}$) end-members (with 95% confidence intervals) of $Fm_{bio} = 0.847 \pm 0.084$ and $TOC_{petro} = 0.108 \pm 0.045$ wt%. Due to concerns about incomplete carbonate removal from differences in decarbonation procedures (see Supplemental Fig.

S4), we also fit Fm to TOC leaving out points with $\delta^{13}C$ greater than -20‰, generating $Fm_{bio} = 0.892 \pm 0.118$ and $TOC_{petro} = 0.213 \pm 0.181$ wt%. These values are within uncertainty of the values from fitting with all the data points. Fitting to TOC:TN weight ratios yielded $Fm_{bio} = 0.956 \pm 0.126$ and leaving out high $\delta^{13}C$ gave $Fm_{bio} = 0.953 \pm 0.217$; these values agreed with the estimates from the TOC fits but with more uncertainty. Therefore, we present the fit of the complete Fm and TOC datasets in Fig. 4d. A fitted value of $Fm_{bio} < 1$ indicated the presence of an aged biospheric endmember in floodplain sediment. This result

was within the range of our measurements of cm-scale wood fragments from within sediment bank samples and cores (Fm ranging from $0.2319 \pm 0.0015$ to $0.9843 \pm 0.0027$, equivalent to radiocarbon ages of $11,750 \pm 55$ to $125 \pm 20$ yr BP); since woody debris does not contain any petrogenic OC, its Fm directly reflects storage and aging in these deposits.





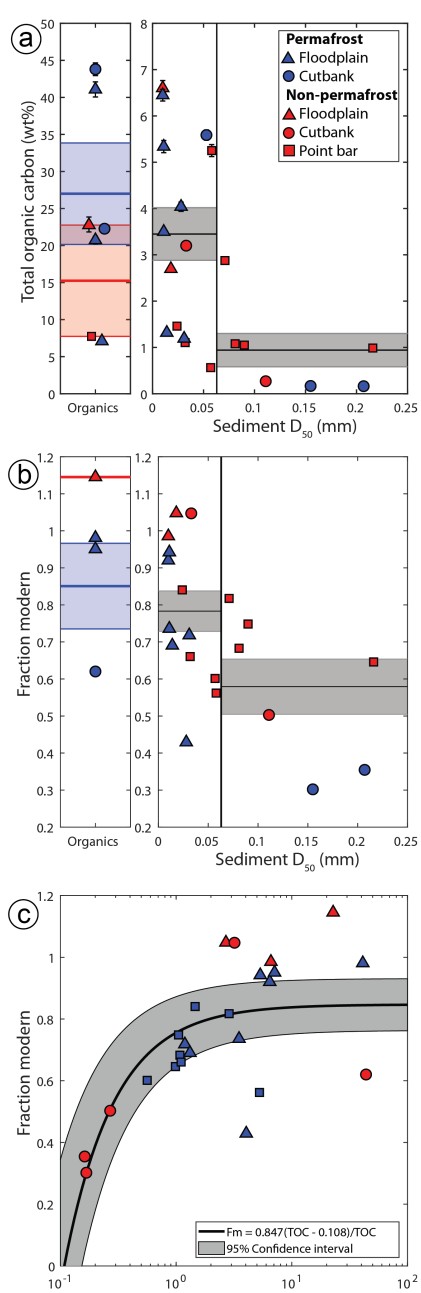



**Figure 4: Floodplain sediment geochemistry results. (a) Total organic carbon versus median sediment grain size, with organic horizons split into ice-rich permafrost peat and non-permafrost topsoil, with 1SD error bars. The horizontal lines indicate the mean and shaded region the standard error of the mean for the peat (n=5, blue shading), topsoil (n=2, red shading), silt ($D_{50}$ < 0.63 mm, n=14, grey shading), and sand ($D_{50}$ > 0.63 mm, n=7, grey shading) grain size classes. (b) Radiocarbon composition (reported as fraction modern, Fm) versus median grain size, with 1SD error bars and shaded regions indicating the mean and standard error of**
**the mean for peat (n=3), topsoil (n=1), silt (n=13), and sand (n=7). (c) Sediment sample fraction modern ($Fm_{meas}$) plotted against TOC ($TOC_{meas}$) and fit using Eq. (7) to calculate end-members for biospheric radiocarbon fraction modern ($Fm_{bio}$) and petrogenic organic carbon content ($TOC_{petro}$).**

To compare the $Fm_{bio}$ of sediment samples from different terrain types, we assumed a constant $TOC_{petro}$ and calculated $Fm_{bio}$

for each sediment sample based on measured sediment Fm and TOC (Fig. 5c), propagating through 95% confidence intervals from fitting the petrogenic end-member (Fig. 4d). We found that all grain size classes had the same calculated $Fm_{bio}$ within uncertainty, indicating that the relationship between Fm and grain size was due to changes in $f_{petro}$ (Fig. 5c & 4d). All terrain types contained samples with $Fm_{bio}$ < 1, implying that all floodplain $TOC_{bio}$ had been aged independently of the terrain type, including the presence or absence of permafrost.

## 5 Analysis: organic carbon cycling by river meandering

### 5.1 Mass balance model for a meandering river

To evaluate particulate OC fluxes into and out of the Koyukuk River, we used a mass-balance model applicable to single-threaded, meandering rivers (Fig. 1a), neglecting fluxes due to dissolved OC and woody debris. Our model includes vertical variations in floodplain structure and their corresponding OC stocks, following similar floodplain-river exchange models

(Lauer and Parker, 2008). While other models exist that incorporate more complex boundary conditions and sediment tracking (Lauer and Parker, 2008; Malmon et al., 2003; Lauer and Willenbring, 2010), we sought a simple framework in order to use our field data to constrain carbon fluxes. We considered POC fluxes into the river due to cutbank erosion ($F_{CB}$; kg yr$^{-1}$), and out of the river due to POC being deposited in point bars ($F_{PB}$; kg yr$^{-1}$) or overbank deposits ($F_{OB}$; kg yr$^{-1}$) or oxidized during transport and released to the atmosphere as $CO_2$ ($F_{OX}$; kg yr$^{-1}$; Fig. 1b) (Striegl et al., 2012; Denfeld et al., 2013; Serikova et

al., 2018).

$$\frac{d(POC)}{dt} = F_{CB} - F_{PB} - F_{OB} - F_{OX}, \quad (6)$$

These fluxes were calculated using the mean lateral migration rate over 83 km river length comprising 8 meander bends in our study (Fig. 2). We averaged over a long river length in an attempt to capture the characteristic sediment transport distances between depositional events (Pizzuto et al., 2014) and variation in local erosion rate due to channel curvature (Sylvester et al.,

2019; Howard and Knutson, 1984) and the formation of cutoffs and oxbow lakes. We calculated the mean bank erosion rate by first averaging the area of floodplain eroded (1.60 km$^2$) and accreted (1.85 km$^2$) from previously published erosion masks generated using Landsat imagery (Rowland et al., 2019). Dividing this area by the length of the channel reach centerline



(82.823 km) and the measurement interval for the erosion masks (2018-1978) resulting in a mean lateral migration rate of of 0.52 m yr$^{-1}$.


To quantify POC fluxes due to channel migration, we approximated the flux into the river due to cutbank erosion as $F_{CB} = L \times E \times C_{CB}$, where $L$ is a representative river reach length (1 m); $E$ is the bank erosion rate (0.52 m yr$^{-1}$); and $C_{CB}$ is the carbon content of the cutbank (kgOC m$^{-2}$), defined by:

$$C_{CB} = \sum_{i=1}^{n} \rho_i \times H_i \times C_i (1 - M_{H2O,i}).$$ 

(7)

We accounted for $n$ stratigraphic units (e.g., sand and mud beds) that may have different carbon contents, where $\rho_i$ is the mean unit bulk density (kg wet sediment per m$^3$), $H_i$ is the unit thickness (m), $C_i$ is its total OC by mass (kgOC per kg dry sediment of each unit) and $M_{H2O,i}$ is the mass fraction of water in the unit (kg H$_2$O per kg wet sediment of each unit). The point bar carbon flux was similarly calculated using $F_{PB} = L \times E \times C_{PB}$, where $C_{PB}$ is the carbon content of the point bar (kgOC m$^{-2}$).

Floodplain sediment OC stocks were calculated using trends in TOC and Fm with median sample grain size (Fig. 4a-b). The measured stratigraphic sections were divided into 4 units (Supplemental Fig. S4): sand ($D_{50} > 63$ μm), mud ($D_{50} < 63$ μm), topsoil (organic horizons overlying non-permafrost sediment) and peat (organic horizons overlying permafrost). We used these groups and calculated the $C_{CB}$ or $C_{PB}$ for each sampled location using Eq. (7), with stratigraphic unit height ($H_i$) taken from the stratigraphic column at each location (Supplemental Fig. S3). We used the mean TOC ($C_i$) and mass fraction of water

($M_{H2O,i}$) and Gaussian error propagation of 1 standard deviation as the value of the sand, mud, topsoil, and peat stratigraphic units (Supplemental Tables S2-S4). We used a constant mean bulk density ($\rho_i$) across all stratigraphic units, because bulk densities measured from core samples for mineral (mean±SD of 989±323 kg m$^{-3}$, n=7) and organic (905±49 kg m$^{-3}$, n=2) horizons were the same within uncertainty (Supplemental Table S2).

Total OC measurements (Fig. 4a; Supplemental Fig. S6-7) were averaged for each grain size class and integrated over 1 m depth below the surface (Fig. 5a) and a bank thickness equivalent to the bankfull depth (12.4 m; Fig. 5b). Measurement and sampling were only possible on the exposed section of the riverbank, above the water table. Exposed sections represented 7-47% of total bank height (as measured from channel thalweg to bank top), whereas the rest was submerged and inaccessible. We assumed all sediment below the base of our stratigraphic sections consisted of sand, which was supported by our

measurements of grab samples of the active channel and cores of the floodplain beyond 2 m depth (Supplemental Fig. S3), and was consistent with downward-coarsening trends widely observed in meandering rivers and their deposits (Supplemental Tables S3 & S4) (Miall, 2013). To evaluate the sensitivity of our results to this assumed stratigraphy, we also summed the carbon content to 1 m depth below the ground surface. These 1 m OC stocks also allowed us to compare to previous work, as soil OC stocks are commonly calculated for the upper 1 m of the soil column (Hugelius et al., 2014).

Earth **Surface**
**Dynamics**
Discussions




**Figure 5: Carbon cycling due to river meandering. (a)** Total organic carbon (OC) in each stratigraphic column integrated to 1 m below surface, with unmeasured section assumed to be sand, and horizontal lines indicate the mean and shaded regions 1SD for the complete dataset. **(b)** Total organic carbon in each stratigraphic column integrated to mean channel depth (12.4 m) using same assumptions and uncertainty. **(c)** Calculated fraction modern ($Fm_{bio}$) of the biospheric organic carbon component versus grain size.

Shaded regions are the mean and standard error of the mean for each grain size class. **(d)** The net OC flux due to channel migration is comparable to floodplain net ecological productivity (NEP), and both are zero within uncertainty. The net flux of OC into the river due to erosion of cutbanks and out of the river due to sediment deposition in point bars in the Koyukuk River is calculated as the mean OC stock for each landform (with ±1SD OC stock uncertainty for that landform) multiplied by an average channel migration rate for a 1 m downstream section of riverbank. The cutbank and point bar fluxes are differenced to calculate the net

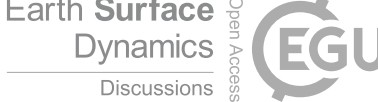

**bank erosion flux. Floodplain NEP is calculated for a 10 km wide, 1 m downstream distance section of floodplain using previously reported regional NEP and uncertainties (Potter et al., 2013).**

### 5.2 Floodplain organic carbon stocks

Estimated permafrost cutbank and floodplain OC stocks integrated to 1 m depth were 31.1±9.8 kgOC m$^{-2}$ (mean ±1SD of OC stocks; n=14), while non-permafrost cutbanks, floodplains and point bars contained 23.3±4.8 kgOC m$^{-2}$ (n=10) (Fig. 5a). The

Mann-Whitney U-test found that OC stocks in permafrost and non-permafrost deposits had similar organic content distributions ($p$= 0.1669). Grouping results by terrain type, permafrost and non-permafrost cutbanks had 30.2±9.2 kgOC m$^{-2}$ (n=11), permafrost and non-permafrost floodplains had 28.8±8.3 kgOC m$^{-2}$ (n=9), and non-permafrost point bars had 19.4±5.2 kgOC m$^{-2}$ (n=4). The Mann-Whitney U-test could not reject the null hypothesis of cutbank and floodplain OC stocks being drawn from the same distribution at 5% confidence ($p$= 0.7891), but the test found weak evidence for point bars having

distinctly lower OC stocks ($p$= 0.0503 for floodplains versus point bars, $p$= 0.0601 for point bars versus cutbanks). Therefore, floodplains and cutbanks generally have higher OC stocks in their upper 1 m of sediment than point bars, but we did not observe a significant difference in 1 m OC stocks between permafrost and non-permafrost deposits (Fig. 5a).

Estimated permafrost cutbank and floodplain OC stocks integrated over the channel depth were 125.1±14.9 kgOC m$^{-2}$ (mean

±1SD of OC stocks; n=14), while non-permafrost cutbanks, floodplains and point bars contained 116.1±11.4 kgOC m$^{-2}$ (n=10) (Fig. 5b). The Mann-Whitney U-test could not reject the null hypothesis that OC stocks in permafrost and non-permafrost deposits had the same organic content distributions ($p$= 0.3641). Grouping results by terrain type, permafrost and non-permafrost cutbanks had 125.3±13.1 kgOC m$^{-2}$ (n=11), permafrost and non-permafrost floodplains had 121.0±13.5 kgOC m$^{-2}$ (n=9), and non-permafrost point bars had 114.0±15.7 kgOC m$^{-2}$ (n=4). Again, the Mann-Whitney U-test could not reject the

null hypothesis of all landform OC stocks being drawn from the same distribution at 5% confidence ($p$= 0.3619 for floodplains versus cutbanks, $p$= 0.8252 for floodplains versus point bars, $p$= 0.2799 for point bars versus cutbanks). Therefore, the distribution of OC stocks integrated to channel depth for cutbanks was indistinguishable from the distribution of measured stocks of newly deposited point bars (Fig. 5b).

### 5.3 Carbon fluxes from river meandering

Using OC stocks integrated to channel depth, we estimated fluxes of POC due to bank erosion as $F_{CB}$=222±25 kgOC yr$^{-1}$ and due to point bar deposition as $F_{PB}$=202±28 kgOC yr$^{-1}$ (Fig. 5d). This result means that that OC fluxes due to bank erosion and bar deposition were equal within the uncertainty of our calculations if we only consider fluxes due to cutbank erosion ($F_{CB}$) and point bar deposition ($F_{PB}$).

We used radiocarbon measurements to evaluate if (1) the OC being eroded from cutbanks was oxidized during transport ($F_{OX}$), (2) the eroded OC was re-deposited in bars via lateral accretion ($F_{PB}$) or overbank deposits ($F_{OB}$), or (3) new biospheric OC



was being added to point bars and floodplains by vegetation growth after sediment deposition. Similar to TOC and TN, Fm displayed a trend of higher values for finer grain sizes—a pattern consistent with prior findings that reflects the greater petrogenic OC contribution in coarser material (Hilton et al., 2015; Galy et al., 2007). When sediment radiocarbon content was

corrected for the petrogenic contribution, $Fm_{bio}$ did not exhibit a grain-size dependence but did show evidence for *in situ* biospheric production. This implies that biospheric OC had a similar Fm for all grain sizes, but that fine sediment tended to contain a higher $Fm_{bio}$, potentially due to being located in upper sedimentary strata.

Our mass-balance calculation and aged $Fm_{bio}$ beingpresent in newly deposited point bars both support that a significant fraction

of OC eroded from cutbanks is re-deposited in the floodplain and not oxidized during transport. In addition to point bar deposition, OC could be lost from the river via overbank deposition ($F_{OB}$). In this case, one would expect the carbon stocks to increase on floodplain surfaces of increasing age due to the deposition of silt units near the surface. Our measurements did indicate a slight increase in 1 m OC stocks between recently deposited point bars and floodplain inferred to be older based on their distance to the river (Fig. 5a), but they did not show a significant increase in OC stock when integrated to channel depth

(Fig. 5b). One possible explanation could be that $F_{OB}$ is substantial, but that this carbon has been remineralized and lost to the atmosphere. To constrain the frequency of overbank flooding along the Koyukuk River near Huslia, we examined the Landsat image record and did not find instances of overbank flooding. Ice jams, where floating ices piles up and causes high waters during spring break up along Arctic rivers, occurred only four times near Huslia from 1967 – 2019, and in these cases, overbank flooding did not occur (White and Eames, 1999). Therefore, historical records suggest that sediment fluxes due to overbank

sediment deposition are relatively minor compared to fluxes due to channel migration. Our stratigraphic observations showing the similar thickness of capping silt units in floodplain stratigraphy (with a mean of 1.29 m for cutbank, 0.92 m for floodplain, and 1.55 m for point bar samples; Supplemental Table S4), and the low mass fraction of siliciclastic sediment in organic horizons (based on high mass fraction TOC; Fig. 4a) also indicated that overbank deposition of sediment on the distal floodplain is relatively small.


Rather than additional OC from overbank flows, floodplains do appear to accumulate additional OC from *in situ* biomass production. We observed an increase in organic horizon thickness, from a mean of 0.06 m in point bars to 0.45 m in cutbank and 0.44 m in floodplain deposits, primarily driven by increasing thickness of peat horizons (Supplemental Table S4). The increase in organic horizon thickness can explain the cutbank and floodplain OC stocks summed to 1 m depth being slightly

higher than the point bar 1 m OC stocks. To quantitatively assess the fraction of OC summed to channel depth produced by the biosphere *in situ*, we used a linear mixing model with a topsoil sample (Fm = 1.1507±0.0781) as an end-member for *in situ* biomass and the lowest measured Fm in cutbank woody debris (Fm = 0.2319±0.00152) as an end-member for OC that has been transported and re-deposited using Eq. (5) (Scheingross et al., 2021). Calculated *in situ* biospheric inputs were significant—we estimated that point bars have 58% (37-84%, n=8), floodplains have 69% (23-94%, n=10), and cutbanks have

72% (42-93%, n=5) of sediment TOC produced *in situ* (reported as mean with range of $f_{bio,is}$ and number of sediment samples


Earth **Surface** Dynamics
Discussions
EGU

in parentheses) (Supplemental Table S6). Since OC stocks summed to channel depth were statistically similar between landforms, we expected that there was some oxidation of modern, labile OC during fluvial transport that was replaced after sediment is deposited in a point bar by *in situ* biomass production. In spite of significant *in situ* biospheric OC input, we found that between one-quarter to one-half of point bar OC has been eroded from upstream and subsequently re-deposited, providing

a reservoir of OC that has been aged by sediment storage along the Koyukuk River.

## 6 Discussion

Our mass-balance model indicated that channel migration generated substantial fluxes of OC into the river (>200 kgOC yr$^{-1}$ m$^{-1}$ from cutbank erosion). If we assumed that all OC in point bars was deposited with river sediment, the calculated OC fluxes due to bank erosion and bar deposition balanced each other within uncertainty (Fig. 5d). However, our radiocarbon analyses

indicated that over half of the biospheric OC in point bars was fixed after deposition by local vegetation. This was reflected in slightly higher 1 m OC stocks in cutbanks and floodplain deposits versus point bars. If we instead assumed that around half of OC in eroding cutbanks was oxidized during river transport, based on the estimated contribution of *in situ* production on point bars, we calculated the river must transport downstream or oxidize >100 kgOC yr$^{-1}$ per meter of river reach. For comparison, measurements of floodplain net ecological productivity (NEP)—the rate of OC fixation minus respiration— indicated an

equivalent 10 km wide, 1 m long river reach would emit 12.1±39.9 kgOC yr$^{-1}$ (mean ±1SD) (Potter et al., 2013). Therefore, the high depth (>10 m) and migration rates (0.52 m yr$^{-1}$) of the Koyukuk River allow fluxes due to bank erosion and deposition to exceed floodplain NEP, despite the far smaller land area of erosion and deposition along the riverbanks compared to the expansive floodplain. A significant oxidation flux during transport ($F_{OX}$) agrees with sparse observations of very high observed excess dissolved $CO_2$ and methane in Koyukuk river water (Striegl et al., 2012). However, significant work remains to

understand the partitioning of OC loss between the dissolved and particulate loads, as well as between petrogenic versus biospheric POC, particularly since DOC concentration and lability varies seasonally in the headwaters of the Koyukuk (O'Donnell et al., 2010).

Our results indicated less variability in OC stocks across the Koyukuk river floodplain than previous work by Lininger et al.

(2019), who found significant variations in OC stocks between geomorphic units in the Yukon Flats. Lininger et al. (2019) report OC stocks to a depth of 1 m along the Yukon River and its tributaries and extrapolated the deepest measured mineral OC concentrations to 1 m based on similar OC content in a few samples taken at depth along cutbanks. Similar to their results, we found that newly deposited point bars without a thick organic horizon had slightly lower OC stocks for the upper 1 m of sediment. Our results also agree with Lininger et al. (2019) that the coarser sediment fraction contributes significant OC and

that floodplain sediments can store OC for thousands of years between riverine transport events. However, we found little variation with geomorphic unit for OC stocks calculated to the channel depth (12.4 m). Though we included organic horizons extending below 1 m, the majority of our OC budget used to calculate fluxes due to channel migration was comprised of the more massive sandy deposits with low OC concentration. These differences point to the importance of river depth relative to



the depth of significant *in situ* biospheric OC input and the grainsize of the floodplain material at depth. We hypothesize that

cutbank and point bar OC stocks will be similar for rivers with coarser sediment and channels much deeper than the active layer and rooting depth of vegetation. In contrast, we expect that OC stocks in floodplains of fine-grained, shallow rivers will have a higher fraction of their OC oxidized after erosion from cutbanks and replaced after deposition in point bars.

The presence of aged biospheric OC in newly deposited, non-permafrost point bars along the Koyukuk River illustrated that

floodplains are important reservoirs of aged OC in sediments both with and without permafrost. Rivers tend to rework younger floodplain deposits faster than older floodplain deposits, and this can yield a heavy-tailed distribution of deposit ages and carbon storage over thousands of years (Torres et al., 2017). Our results supported the idea that a fraction of particulate OC has experienced transient mobilization and deposition, and thus becomes naturally aged during transport through the river-floodplain system. Therefore, particulate OC with old radiocarbon signatures might be attributed to OC storage in floodplains,

and may not be a diagnostic indicator of permafrost thaw. One might expect better preservation of carbon stocks in permafrost deposits. However, our field observations of bank sediment rapidly changing color from gray to orange when exposed to air imply that thawed floodplain sediments are predominantly anoxic, which may reduce rates of organic matter respiration in non-permafrost deposits. When coupled with cold mean annual temperatures, anoxic non-permafrost terrain might be similarly effective as permafrost in preserving and aging biospheric OC stocks (Davidson et al., 2006). Thus, transient storage of

particles in floodplains, potentially for thousands of years (Repasch et al., 2020), may delay or diffuse downstream signals of perturbations to the watershed's carbon cycle before reaching long-term monitoring stations at river mouths or sediment depocenters (McClelland et al., 2016; Holmes et al., 2012).

Climate change is expected to cause a decrease or disappearance of permafrost, which might alter rates of POC oxidation ($F_{OX}$)

and overbank deposition ($F_{OB}$) and ultimately downstream riverine POC fluxes. Permafrost thaw is also hypothesized to increase river lateral migration rates (Costard et al., 2003), although such changes have yet to be systematically documented. For the Koyukuk River, higher channel migration rates should, with all else equal, increase the magnitude of OC fluxes due to erosion and deposition and thereby decrease the residence time and age of OC within the floodplain, but possibly with no net change in OC fluxes from the floodplain to the river. However, if, for example, climate change increases the relative importance

of overbank deposition of OC-rich mud (higher $F_{OB}$) relative to sand bar accretion, then this change would cause a permanent increase in floodplain OC stocks, with associated decreases in OC river fluxes during the transient period of floodplain grainsize fining. In contrast, an increase in channel lateral migration relative to overbank flooding would cause floodplains to become sandier and floodplain OC stocks to decline. Furthermore, climate change is altering flood discharge and frequency (Koch et al., 2013; Vonk et al., 2019; Walvoord and Kurylyk, 2016) as well as sediment supply, often associated with thaw

slumps (Kokelj et al., 2013; Lantz and Kokelj, 2008; Malone et al., 2013; Shakil et al., 2020). Increases in flood magnitude could cause channel widening (Ashmore and Church, 2001; Walvoord and Kurylyk, 2016), which would increase cutbank OC fluxes relative to point bar fluxes ($F_{CB} > F_{PB}$), creating a transient increase in riverine OC flux. We expect that changes in



floodplain hydrology and sedimentation due to climate change will alter downstream particulate OC fluxes and floodplain OC storage along deep, meandering Arctic rivers similar to the Koyukuk. In the process, sediment deposition in river bars should preserve radiocarbon-depleted OC and dampen positive feedbacks due to POC being released from permafrost by riverbank erosion as the climate warms.

## 7 Conclusions

To evaluate the role of riverbank erosion and bar deposition in liberating organic carbon (OC) from permafrost floodplains, we conducted a field campaign along the Koyukuk River in central Alaska, taking samples of riverbank and floodplain sedimentary deposits. Finer bank sediment had a systematically higher TOC and Fm than coarser sands. We combined measurements on individual samples with measured floodplain stratigraphic columns to calculate OC stocks for cutbanks, point bars and floodplain cores summed both to 1 m below the surface and to the river channel depth. We found that cutbanks had slightly higher OC stocks than point bars at shallow depths. However, that OC stocks calculated to river channel depth did not significantly vary between river cutbanks, floodplain and point bars or with the presence or absence of permafrost. Our results indicated that floodplain processes generated an aged biospheric radiocarbon signature that did not vary with grain size, and variations in sediment Fm were primarily due to mixing with a petrogenic end-member. We concluded that approximately one-quarter to one-half of biospheric OC that was eroded from cutbanks was preserved through transport and deposition. The presence of radiocarbon-depleted sediment in non-permafrost deposits indicated that aged POC in Arctic rivers is not a unique indicator for the presence of permafrost. Our results highlighted that Arctic floodplains are significant reservoirs of OC, and their stratigraphic architecture and morphology influence POC fluxes and radiocarbon ages transmitted downstream. Therefore, sediment deposition in river bars should dampen positive feedbacks due to POC being released from permafrost by riverbank erosion as the climate warms.

### Data availability

All datasets are included in the manuscript and supplemental material.

### Author contributions

MPL, WWF, AJW, JCR, GKL, & MMD conceptualized the study; MPL, AJW, JCR & GKL determined the methodology; MMD, GKL, JCR, PCK, AJW, JS, APP, AJC, & MPL collected field data; MMD, GKL, PCK, & AJW assisted with geochemistry; MPL supervised the work; MMD conducted data analysis and wrote the original draft; and all authors contributed to the review and editing of the writing.



**Competing interests**

The authors have the following competing interests: One author is a member of the editorial board of Earth Surface Dynamics. The peer-review process was guided by an independent editor, and the authors have also no other competing interests to declare.

**Acknowledgements**

We thank the Koyukuk-hotana Athabascans of Huslia, First Chief Norman Burgett and the Huslia Tribal Council for access to their land, and U.S. Fish and Wildlife Service (USFWS) – Koyukuk National Wildlife refuge for research permitting and logistical assistance. Shawn Huffman, Alvin Attla, and Virgil Umphenour provided field support and local expertise. We also thank Matthew Kirby for use of the Malvern Mastersizer and assistance with grain size analysis.

We acknowledge financial support from the Caltech Terrestrial Hazards Observation and Reporting Center, Foster and Coco Stanback, and the Linde Family to MPL and WWF; the Caltech Center for Environmental Microbial Interactions to WWF; a Department of Energy Office of Science, Biological and Environmental Research, Earth & Environmental Systems Sciences Division, Subsurface Biogeochemical Research Program Early Career Award  to JCR; the National Defense Science and Engineering Graduate Fellowship for MMD and PCK; and the Fannie and John Hertz Foundation Cohen/Jacobs and Stein
Family Fellowship for PCK.

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
