# Peer review of "Organic carbon burial by river meandering partially offsets bankerosion carbon fluxes in a discontinuous permafrost floodplain"

_Earth Surface Dynamics, 2021_

## Referee Comment (RC1)

*Review of Douglas et al. "Organic carbon burial by river meandering partially offsets bank-erosion carbon fluxes in a discontinuous permafrost floodplain" (esurf-2021-80)*

Synopsis

The primary focus of this study is to constrain the source, age, and flux of carbon that is entrained, transported, and deposited along an Arctic river underlain by discontinuous permafrost (Koyukuk River, Alaska, USA). In particular, the authors collected a suite of samples from eroding cut banks, depositional point bars, and floodplain deposits and compared their grain size, organic carbon content (TOC), and organic carbon $\delta^{13}C$ and $^{14}C$ activity (Fm). They found that finer-grained material is associated with higher TOC and higher Fm, but that carbon stocks are statistically identical in cut banks, point bars, and floodplains. As a consequence of this result, the authors argue that a large fraction of mobilized OC is redeposited and aged during fluvial transport (independent of permafrost cover), rather than being oxidized to $CO_2$.

Overall, I find this study to be highly topical and relevant to an import carbon-cycle question. In particular, I find the combination of techniques taken from organic geochemistry (i.e., $\delta^{13}C$ and Fm) as well as geomorphology/sedimentology (i.e., migration rates, stratigraphic columns, etc.) to be an exciting contribution. That said, there are several statements and calculations that are contradicting, circular, or warrant further clarification—these particularly relate to the interpretation of Fm results. Most importantly: (i) the calculation of individual $Fm_{bio}$ estimates for each sample requires the authors to input $TOC_{petro}$, which is calculated by assuming a constant $Fm_{bio}$ for all samples—this is circular logic. And (ii) the calculations used herein lead to incorrect (artificially high) estimates of the fraction of OC that is produced *in situ*. Still, these issues do not invalidate the main conclusion of the paper, which is that OC stocks in eroding cut banks and deposited point bars are statistically identical, independent of the presence of permafrost.

I detail these points below, along with some minor (line-item) comments. Once these issues have been fully addressed, I support publication of this manuscript in *ESurf*. Please do not hesitate to contact me for further details regarding this review.

Sincerely,

Jordon Hemingway
jordon.hemingway@erdw.ethz.ch

L14-15: "Radiocarbon content" and "radiocarbon abundance" should be replaced by "radiocarbon activity," since this is the unit of currency used in radiocarbon measurements.

L14-15: "TOC" should be "TOC content" or "TOC abundance" or similar.

L75: This should read, "…fluxes of OC into *and out of* the river…"

L85: This is not the correct definition of Fm. This should instead be:

$$Fm \; = \; \frac{A_{S,N}}{0.95 \times A_{Ox,N}},$$

where $A_s$ is the $^{14}C$ *activity* of the sample, $A_{Ox}$ is the $^{14}C$ *activity* of the Ox-I oxalic acid standard, "N" indicates that both activities are normalized for isotope fractionation (to $\delta^{13}C = -25‰$ for the sample and -19‰ for Ox-I), and the scalar 0.95 is added by convention. See Reimer et al. (2004) *Radiocarbon* **46**, 1299–1304.

Additionally, it sounds a bit strange to say "…with low fractions *of* modern radiocarbon…" I suggest rewording to "…with OC containing low radiocarbon activity, expressed as fraction modern (Fm = …"

L109: "…with a constant mass fraction of petrogenic OC." State why $OC_{petro}$ weight percent should be constant—i.e., since it is all eroded from the same bedrock units, independent of grain size.

L111-120: This set of equations (and possibly the subsequent regression and results) is not strictly accurate. If I read this right, Eq. (4) simply reduces to "$Fm_{meas} = Fm_{bio}(1 - f_{petro})$". The authors state that the mass fraction of $OC_{petro}$ is constant (which is reasonable), but this is not the same as $f_{petro}$ being constant! Rather, $f_{petro}$ is the fraction *of total OC* that is petrogenic. This should instead be rewritten as

$$TOC_{meas} = TOC_{bio} + TOC_{petro},$$

and

$$TOC_{meas} \times Fm_{meas} = TOC_{bio} \times Fm_{bio} + TOC_{petro} \times Fm_{petro},$$

such that

$$Fm_{meas} \; = \; Fm_{bio} - TOC_{petro} \times Fm_{bio} \left( \frac{1}{TOC_{meas}} \right).$$

Then, a plot of $TOC_{meas}$ (or $1/TOC_{meas}$) vs. $Fm_{meas}$ yields an intercept that is $Fm_{bio}$ and a slope (or curvature) that is proportional to $TOC_{petro}$, which is the metric that the authors assume to be constant across all samples.

L121: Right, here it is clear that $TOC_{petro}$, not $f_{petro}$, is constant!

L130: Note that this mixing model is only valid if $Fm_{bio,is} > Fm_{bio}$, which might not always be the case due to the decreasing $^{14}C$ activity of the atmosphere since nuclear bomb testing ended in the 1960s. That is, biospheric OC fixed ~60 years ago will have a higher Fm value that *in situ* biospehric OC being fixed today. Incorporation of OC fixed over the past ~60 years would therefore artificially increase the calculated $f_{bio,is}$. These complexities need to be addressed and discussed.

L131: "…$Fm_{bio}$ is the fraction modern of biospheric OC for each sediment sample…" But won't this be the same for all samples, since a regression of Eq. (4) yields only a single $Fm_{bio}$ value for the whole sample set? Then, if $Fm_{bio,is}$ and $Fm_{bio,cb}$ are assumed to be constant, I don't see how Eq. (5) doesn't yield a constant value for $f_{bio,is}$ for all samples.

L134: But is the "oldest woody debris from a cutbank" actually representative of the cutbank end member? Should this not instead be the *average* Fm value of all cutbank samples (or at least the average of the cutbank woody debris samples)? Again, the choice used by the authors here will artificially increase the calculated $f_{bio,is}$ (i.e., since the true Fm of all OC inputted from the cutbank is almost certainly higher than that of the oldest woody debris). This should be discussed.

L189: Which values—LANL or NOSAMS—were used in all subsequent calculations? For some metrics in some samples, this appears to make quite a difference.

L217: This should read "Sediment TOC and radiocarbon Fm varied…" (remove "measurements")

L231-232: Can the authors elaborate on what they mean by "Fitting to TOC:TN weight ratios…"? Wouldn't this yield the (somewhat odd) result of $TOC_{petro}:TN_{meas}$? I'm confused by the motivation or benefit of calculating the regression in this way.

L249-251: But the regression in Fig. 4d requires the assumption of a *constant* $Fm_{bio}$ across all samples in order to calculate $TOC_{petro}$. So, how can this result then be used to estimate variable $Fm_{bio}$ values for each sediment sample? This seems circular and will, by definition, yield estimated $Fm_{bio}$ values that are the same across all grain size classes.

L273: Remove one of the redundant "of" instances

Fig. 4: Again, I'm not really sure $Fm_{bio}$ can be calculated for each individual sample, when the calculated inputs for this calculation (i.e., $TOC_{petro}$) *require* the assumption of constant $Fm_{bio}$ across all samples (i.e., Eq. 4). This seems circular to me.

L349-350: Clarify by changing to "…reflects the greater *proportional* petrogenic OC contribution in coarser material"

L350-352: How can biospheric OC simultaneously have a "similar Fm for all grain sizes" but "fine sediment … contain higher $Fm_{bio}$"? These statements are contradicting. I'm also not convinced by the evidence for *in situ* biospheric production (as detailed above and below).

L354: Space between "being present"

L376-380: As mentioned in my above comments (L130, L134), this calculation will lead to an artificially high estimate of $f_{bio,is}$. Rather, the average $Fm_{bio}$ for cut bank inputs—which is more representative for the calculation of $f_{bio,is}$—is much higher than 0.2319±0.00152.

This again comes back to the regression from Eq. (4) used to calculate a constant $Fm_{bio}$ and $TOC_{petro}$ across the entire sample set. Given this calculation, all samples will by definition have the same calculated $Fm_{bio}$ and thus there is no "room" to oxidize and replace some $TOC_{bio}$ with $^{14}C$-enriched *in situ* material.

Furthermore, how is it possible that "cut banks have 72% of sediment TOC produced *in situ*" (L380), but cut bank $Fm_{bio}$ ($Fm_{bio,cb}$) is also used as the non-*in situ* end-member in Eq. (5) (L130)? According to Eq. 5, shouldn't cut banks have 0% *in situ* TOC by definition? This needs to be clarified and discussed further.

I am also now somewhat confused as to what specifically the authors mean by "*in situ*" production. Does this refer only to TOC produced during transport within the river itself (I think this is the case), or does this refer to any non-permafrost $TOC_{bio}$ (which I think is what is implied by using the oldest woody debris for the non-*in situ* end member Fm value)? If the former, then why would the authors expect their results here to deviate so far from previous estimates (e.g., Scheingross et al., 2021), which conclude that there is very limited oxidation and *in situ* production during fluvial transport?

I think the assumptions and calculations used to calculate $f_{bio,is}$ need to be critically reevaluated. This also warrants updating some discussion points later in the manuscript (e.g., L390, L398, L461)

---

## Referee Comment (RC2)

**Review of Douglas et al., 2021, Organic carbon burial by river meandering partially offsets bank-erosion carbon fluxes in a discontinuous permafrost floodplain. Submitted to *Earth Surface Dynamics*.**

**Summary**

This manuscript explores the effect of riverbank erosion on the fate of permafrost organic carbon, specifically examining the balance between removal of OC from permafrost cutbanks and deposition of OC onto point bars in the Koyukuk River, Alaska. The key question being asked is whether OC mobilized from ancient permafrost deposits is oxidized to $CO_2$ within river systems or is it quickly re-deposited and buried in floodplains. This study is timely because the effects of climate change on the arctic carbon sink cannot yet be predicted, and studies like this will enable us to predict changes in permafrost carbon stocks as river channel migration rates accelerate in the future. This is a relevant scientific question within the scope of *ESurf*.

The authors quantified OC stocks and used a 1-D mass-balance model to quantify net fluxes of OC from cutbanks, to point bars, and to downstream transport. The estimation of carbon stocks and the balance of OC fluxes in the river system appear robust, however, the estimation of OC produced by floodplain vegetation needs to be revisited. The authors should also revisit the biospheric fraction modern values calculated for individual samples. These issues do not appear to affect the overall message of the manuscript, which is that biospheric OC production on point bars is sufficient to offset any OC lost to downstream transport or OC oxidation in the floodplain. However, there are flaws in their calculations that need to be corrected.

Overall, I think this is a nice study that will be of interest to readers of *ESurf*. However, details regarding some measurements and calculations need to be revised before this manuscript can be accepted. Therefore, I support publication after the authors address the concerns detailed below.

**Detailed comments**

L15: change *radiocarbon abundance* to *radiocarbon activity*

What is the timescale over which sediment is deposited onto the point bar?

What is the role of petrogenic OC? Is it possible that petrogenic OC gets preferentially deposited on point bars because it is associated with the denser mineral fraction, while aged biospheric organic matter is transported downstream due to its lower density?

L85: This equation for fraction modern is incorrect. See Jordan Hemingway's review for reporting the correct equation. Equation 1 in Reimer et al. (2004) is the full equation. Other useful references for fraction modern reporting are: Stuiver and Polach, 1977; Donahue et al., 1990.

I also recommend rephrasing "low fractions of modern radiocarbon" to "low radiocarbon activity"

L87: "…low Fm *values* inherited…"

L91: Avoid saying "low or high Fm carbon." Alternatively, I would recommend saying "organic carbon with low Fm values", or "radiocarbon depleted OC".

L104: TOC "concentrations" and Fm "values." Fm is not a measurement, but a value calculated from measurements of 14C and 12C.

L107: In the parentheses, do you mean to say $OC_{petro}$ and $OC_{bio}$, rather than $TOC_{petro}$ and $TOC_{bio}$?

L108: TOC "concentrations"

L110: Figure 4d does not exist.

L110: Constant $f_{petro}$ or $TOC_{petro}$? Make sure the appropriate terms are used throughout the text.

L127: Briefly define what you mean by *in situ*.

L134: Why choose the oldest woody debris as the end-member? The permafrost OC is a mixture of organic matter varying in 14C activity, and as such should be represented by a mean value for ancient permafrost OC samples/woody debris samples. By using the oldest-value as the end-member, the mixing model will result in overestimated proportions of $OC_{bio,is}$.

L157: 12.4 m? Need units here.

Section 3.1 Field sampling methods – it's not clear why the paragraphs are broken up like this. I would combine the ADCP depth with the channel migration rate calculations and move the digging/coring sampling method immediately after the categorization of permafrost vs. non-permafrost.

L167: SIPRE "corer"

L170-180: Were the samples not ground to a fine powder prior to EA and EA-IRMS analyses? I worry that the samples were not thoroughly homogenized prior to analysis. This is particularly important for permafrost cutbank samples that contain both mineral-associated OC and particulate plant debris.
Additionally, TOC and TN concentrations would be measured on the elemental analyzer, and stable carbon and nitrogen isotopes would be measured on the EA coupled to the IRMS. Please clarify that both concentrations and isotopic compositions were measured and on which instruments.
I also suggest moving the details about the measurement calibration standards to the supplement, but it is also fine if you leave it in.

L187: It seems confusing that you had TOC concentrations and stable isotope values measured at two different labs. Please add a sentence before L170 that briefly describes why samples were acid-treated using two different methods and then measured at two different facilities. Were the duplicate EA and EA-IRMS analyses performed on all samples? Additionally, please explain why you used the LANL TOC concentrations for the TOCmeas values.

L196: What was the percent concentration of Calgon solution?

L197: Were samples split twice? L191 says samples were split using a riffle splitter.

L199-200: Is it necessary to say that sand card grain size approximations were made? If you only use the laser diffraction particle size data, then the non-quantitative field observations seem irrelevant.

L217: TOC "content" or "concentrations"

Figure 4: Using the term "organics" is somewhat misleading on the plots. I suggest changing to "organic horizon." It would also be helpful if the shaded regions were labeled with blue text, red text, etc. for the material they represent.

L222: yielded lower "Fm values"

L225: to calculate the "proportions" or the "radiocarbon activity" of petrogenic and biospheric end-members?

Why must there only be two end-members? From my perspective, there is a petrogenic end-member, a permafrost-derived aged biospheric end-member, and a modern biospheric end-member. Because you sampled permafrost OC from cutbanks and soil organic horizon OC, you should be able to sufficiently characterize these three end-members.

L229-230: Supplemental figure S4 contains field photos. I think you want to reference Fig. S5. I'm confused about the exclusion of >-20 per mil because all the d13C values on those plots are < -23 per mil – did you exclude the elevated d13C samples from Figures S5 and S6 as well? And it looks like the samples with higher TOC content do not have elevated d13C values, so I don't think incomplete carbonate removal is a concern. However, it is interesting that the NOSAMS d13C values were often significantly higher than the LANL d13C values. I worry that because the radiocarbon measurements were made on the same aliquots as these elevated d13C measurements, then the radiocarbon activities would also be influenced by any remaining carbonate.

L229: Fig. 4c

L252: If $f_{petro}$ varies across samples, but you are still using the regression to calculate $Fm_{bio}$ for individual samples, you will end up calculating similar $Fm_{bio}$ values for all samples. You would need a different end-member mixing model approach to calculate $f_{bio}$ for individual samples.

L280: It might be helpful to write out the assumption that $M_{H2O} + M_{dry} = 1$.

L340: How do these OC fluxes compare with the OC flux exported by the river? A more thorough mass-balance would account for the sediment flux in the river and determine how much is deposited onto point bars over the ~80m reach studied. Although, it may not be necessary if the main point is that modern biospheric production and soil development on point bars is the primary mechanism of balancing OC stocks between cutbanks and point bars.

L354: need space between *being* and *present*

L380: I think the authors need to re-calculate the percentages of *in situ* biospheric OC in the samples. It seems highly unlikely that cutbanks have ~72% of sediment OC produced *in situ*

given what we know about permafrost carbon stocks. I think this can be corrected by using a permafrost OC cutbank end-member with a higher mean Fm value.

L382: Some oxidation of modern, labile OC? Or rather, oxidation of labile ancient permafrost-derived OC?

L414: "biospheric OC *production*?"

L455: TOC content and Fm values

L459:

L460: The authors should reconsider the calculations of $Fm_{bio}$ for individual samples and not make a comparison across grain sizes using those estimated values.

**Figures**

Figure 1: Is it true that channel width has been remaining constant over the last few decades? Based on Figure S2, it appears that there is a minor shift to increased channel widths, and I wonder if mean values would be better suited to characterize this measurement. Mean values would only be appropriate if the channel width is measured continuously along the study reach.

Figure 4c: Should the blue squares be blue circles?

Figure S1: I recommend using different colors or shapes to distinguish between core, bank, and pit samples, like that used in Figure 2.

**References Cited**

Donahue, D. J., Linick, T. W., & Jull, A. T. (1990). Isotope-ratio and background corrections for accelerator mass spectrometry radiocarbon measurements. *Radiocarbon*, *32*(2), 135-142.

Reimer, P. J., Brown, T. A., & Reimer, R. W. (2004). Discussion: reporting and calibration of post-bomb 14C data. *Radiocarbon*, *46*(3), 1299-1304.

Stuiver, M., & Polach, H. A. (1977). Discussion reporting of 14C data. *Radiocarbon*, *19*(3), 355-363.

---

## Author Comment (AC1)

[revised manuscript text omitted]

935 **Fig. S2: Koyukuk River width.** Probability distribution of width of channel masks generated from Landsat 30 m imagery, with widths calculated at each pixel along the channel centerline for the reach of the Koyukuk River pictured in Fig. S1 (Rowland et al., 2019). The reach maintained a roughly constant channel width over the Landsat record, from 173±43 m (median ± 1SD) in 1978 to 179±42 m in 2018, supporting our assumption that cutbank erosion and point bar deposition occur at the same rate.

[Figure]

**Cutbanks**

[Figure]

**Fig. S3: Measured stratigraphic sections grouped by location on river floodplain.** The river channel is eroding its cutbanks and depositing sediment on its point bars, which accrete to form the floodplain as the channel continues to migrate laterally. Note that deposits were generally sandy greater than 2 m depth below the floodplain surface, and that organic horizon thickness at the ground surface varies, though lenses of organic-rich sediment were prevalent meters below the surface. The active layer was shallower in permafrost units containing thick layers of peat, while locations without permafrost contained plant roots extending meters farther below the ground surface and lacked thermoerosional niches. Thicknesses of stratigraphic units were tabulated for each section in Table S4.

940

945

[Figure]

Fig. S4: Field photos of floodplain stratigraphic facies. (a) Permafrost sand in Bank 9 with 2 inch PVC pipe (outer diameter is 6.0 cm) installed in the bank for scale. (b) Pit dug in non-permafrost ground with root-rich topsoil overlying silt at Core 5. (c) Non-permafrost sandy deposits on a point bar beach at Pit 9. (d) Permafrost silt containing ice lenses in Core 4. (e) Overhung cutbank from Bank 9, with a layer of peat overlying an ice wedge surrounded by grey, frozen silt with slump blocks and intraclasts of thawed peat and silt forming a slope that shields the bank.

950

[Figure]

**Fig. S5: Organic carbon measurement comparison, with samples with ¹³C values > -20‰ excluded**. TOC and OC stable isotopes were measured at both NOSAMS and Los Alamos National Lab (LANL), with NOSAMS generally showing a slightly higher TOC and δ¹³C. We attribute these differences to decarbonation procedures : using HCl in solution for the NOSAMS measurements and fumigation with HCl for the LANL measurements (see Sect. 3). All plots in the main text and supplemental materials use the LANL TOC and δ¹³C values with NOSAMS radiocarbon measurements. (a) NOSAMS versus LANL TOC measurements, with error bars showing 1SD analytical uncertainty. (b) Zoomed in plot of shaded region in plot (a). (c) NOSAMS versus LANL OC stable isotope measurements, reported as per mille (‰) relative to VPDB with error bars showing 1SD analytical uncertainty.

955

960

[Figure]

**Fig. S6: Sediment OC characteristics, with samples with [13]C values > -20‰ excluded. Stable organic carbon isotopes displayed no trends with (a) inverse total organic carbon (TOC) or (b) radiocarbon fraction modern (Fm). Sediment δ[13]C values spanned the range previously reported in peat and woody debris (from -23.2±0.2 ‰ to -28.6±0.2 ‰) on the Koyukuk River floodplain near its confluence with the Yukon River (O'Donnell et al., 2012). Stable organic carbon isotope values also incorporated a petrogenic end-member, and kerogen-rich sedimentary rocks in the Brooks Range had δ[13]C ranging from -27.23±0.1 ‰ to -30.75±0.1 ‰ (Johnson et al., 2015). Measured δ[13]C values are reported in units of per mille (‰) relative to the VPDB~~, with x and y error bars showing. (c) Fm decreased for high values of 1/TOC but spanned a wide range for low values of 1/TOC, which we interpret as reflecting mixing between modern biospheric, aged biospheric, and petrogenic end-members. Sample x and y error bars show 1SD analytical uncertainty.**

[Figure]

**Fig. S7: Sediment total nitrogen.** (a) Total nitrogen (TN) versus total organic carbon (TOC) measured as dry weight % of samples, with error bars showing 1SD analytical uncertainty. (b) Radiocarbon fraction modern values versus molar ratio of organic carbon to nitrogen, with error bars showing 1SD analytical uncertainty.

**Table S1: Sample site locations and characteristics.**

| Sample site | Landform | Latitude (°) | Longitude (°) | Frozen ground type |
|---|---|---|---|---|
| Bank 1 | Cutbank | 65.78014 | -156.43661 | Permafrost |
| Bank 2 | Cutbank | 65.76493 | -156.49031 | Non-permafrost |
| Bank 3 | Cutbank | 65.76519 | -156.48964 | Non-permafrost |
| Bank 4 | Cutbank | 65.75232 | -156.50511 | Permafrost |
| Bank 5 | Cutbank | 65.75232 | -156.50511 | Permafrost |
| Bank 6 | Cutbank | 65.75232 | -156.50511 | Permafrost |
| Bank 7 | Cutbank | 65.66093 | -156.45087 | Non-permafrost |
| Bank 8 | Cutbank | 65.66126 | -156.44711 | Non-permafrost |
| Bank 9 | Cutbank | 65.70265 | -156.40977 | Permafrost |
| Bank 10 | Cutbank | 65.61942 | -156.48534 | Permafrost |
| Bank 11 | Cutbank | 65.62931 | -156.46198 | Non-permafrost |
| Bank 12 | Cutbank | 65.64022 | -156.50949 | Non-permafrost |
| Bank 13 | Cutbank | 65.87132 | -156.26283 | Permafrost |
| Bank 14 | Cutbank | 65.70153 | -156.40353 | Permafrost |
| Core 1 | Floodplain | 65.78014 | -156.43661 | Permafrost |
| Core 2 | Floodplain | 65.76521 | -156.49049 | Non-permafrost |
| Core 3 | Floodplain | 65.72090 | -156.37178 | Permafrost |
| Core 4 | Floodplain | 65.73519 | -156.38866 | Permafrost |
| Core 5 | Floodplain | 65.67904 | -156.61163 | Non-permafrost |
| Core 6 | Floodplain | 65.67158 | -156.58762 | Permafrost |
| Core 7 | Point bar | 65.66046 | -156.43256 | Non-permafrost |
| Core 8 | Floodplain | 65.72552 | -156.20992 | Permafrost |
| Core 9 | Floodplain | 65.71100 | -156.27473 | Permafrost |
| Pit 1 | Point bar | 65.77817 | -156.43370 | Non-permafrost |
| Pit 2 | Point bar | 65.77764 | -156.43364 | Non-permafrost |
| Pit 3 | Point bar | 65.77688 | -156.43394 | Non-permafrost |
| Pit 4 | Point bar | 65.77636 | -156.43342 | Non-permafrost |
| Pit 5 | Point bar | 65.77483 | -156.43354 | Non-permafrost |
| Pit 6 | Point bar | 65.77756 | -156.43381 | Non-permafrost |
| Pit 7 | Floodplain | 65.72083 | -156.37217 | Non-permafrost |
| Pit 8 | Point bar | 65.65986 | -156.43524 | Non-permafrost |

| | | | | | |
|---|---|---|---|---|---|
| Pit 9 | Point bar | 65.65958 | -156.43542 | Non-permafrost |
| Pit 10 | Point bar | 65.66132 | -156.43354 | Non-permafrost |

**Table S2: Sample descriptions and results of laboratory analysis. Starred samples have median grain size ($D_{50}$), TOC, TN, $\delta^{13}C$ and molar TOC/TN ratios previously reported in Douglas et al. (2021).**

985    Attached as file TableS2.csv

**Table S3: Averaged sediment TOC concentrations and constants used in calculations of bank TOC content integrated to channel depth.**

| | Sand | Silt | Peat | Topsoil |
|---|---|---|---|---|
| **$D_{50}$ (mm)** | >0.063 | <0.063 | N/A | N/A |
| **Water content (wt%)** | 18.1±6.1 | 46.6±15.6 | 87.5±7.4 | 62.2±1.0 |
| **TOC (wt%)** | 0.94±0.95 | 3.69±2.25 | 35.20±12.60 | 15.25±10.62 |
| **TOC (kgC/m³)** | 7.49±8.27 | 19.1±14.4 | 42.7±20.0 | 55.9±42.2 |
|  |  |  |  |  |
|  |  |  |  |  |
|  |  |  |  |  |
|  |  |  |  |  |
| **Bulk density (kg/m³)** | 971±283 | | | |
| **Channel Depth (m)** | 12.4 | | | |
| **Migration Rate (m/yr)** | 0.52 | | | |

**Table S4: Calculation of bank TOC,  content integrated to channel depth based on measured**
990    **stratigraphic columns. Note that unmeasured section was assumed to consist of sand based on field observations.**

Attached as file TableS4.csv

**Table S5: Complete grain size distributions measured using laser diffraction tabulated in log-normal bins, with 10th-, 50th- and 90th-percentile grain size reported as $D_{10}$, $D_{50}$, and $D_{90}$.**

995    Attached as file TableS5.csv

|  |  |  |  |
|---|---|---|---|

| | | | |
|---|---|---|---|
| KY18-Bank2-10 | Cutbank sediment | 1.0837±0.0330 | 0.9272±0.0825 |
| KY18-Bank2-230 | Cutbank sediment | 0.8398±0.1015 | 0.6617±0.0906 |
| KY18-Bank9-Peat | Cutbank sediment | 0.6217±0.0173 | 0.4243±0.0355 |
| KY18-Bank9-220 | Cutbank sediment | 1.0631±0.1776 | 0.9047±0.1890 |
| KY18-Bank9-350 | Cutbank sediment | 0.8509±0.1396 | 0.6737±0.1158 |
| KY18-Core1-22-28 | Floodplain sediment | 0.9359±0.0265 | 0.7662±0.0662 |
| KY18-Core1-105-111 | Floodplain sediment | 0.7593±0.0228 | 0.5740±0.0489 |
| KY18-Core2-10-12 | Floodplain sediment | 1.0916±0.0413 | 0.9357±0.0870 |
| KY18-Core2-35-37 | Floodplain sediment | 1.0018±0.0354 | 0.8380±0.0755 |
| KY18-Core3-15-20 | Floodplain sediment | 0.9653±0.0340 | 0.7983±0.0714 |
| KY18-Core3-84-89 | Floodplain sediment | 0.7899±0.0380 | 0.6074±0.0554 |
| KY18-Core4-16-20 | Floodplain sediment | 0.9616±0.0342 | 0.7942±0.0711 |
| KY18-Core4-105-110 | Floodplain sediment | 0.7521±0.0347 | 0.5662±0.0508 |
| *KY18-Core5-15* | *Floodplain sediment* | *1.1507±0.0781* | - |
| KY18-Core7-85-95 | Point bar sediment | 0.7440±0.0551 | 0.5574±0.0563 |
| KY18-Core7-390-400 | Point bar sediment | 0.5736±0.0205 | 0.3719±0.0314 |
| KY18-Core9-33-38 | Floodplain sediment | 0.9837±0.0344 | 0.8183±0.0733 |
| KY18-Core9-169-174 | Floodplain sediment | 0.4409±0.0160 | 0.2275±0.0190 |
| KY18-Pit1-5 | Point bar sediment | 0.7253±0.0383 | 0.5371±0.0491 |
| KY18-Pit2-10 | Point bar sediment | 0.8494±0.0319 | 0.6721±0.0595 |
| KY18-Pit4-20 | Point bar sediment | 0.7588±0.0382 | 0.5735±0.0524 |
| KY18-Pit5-20 | Point bar sediment | 0.9077±0.0401 | 0.7356±0.0679 |
| KY18-Pit6-60 | Point bar sediment | 0.8343±0.0426 | 0.6556±0.0614 |
| KY18-Pit8-40 | Point bar sediment | 0.7323±0.0365 | 0.5446±0.0493 |
| *KY18-Bank14* | *Cutbank woody debris* | *0.2319±0.00152* | - |

1000

---

## Author Comment (AC2)

**Author Comment**
We thank Jordon Hemingway and an anonymous reviewer for their constructive feedback and positive reviews of our manuscript. In response to the concerns both reviewers raised regarding the biospheric and petrogenic OC mixing model, we have substantially revised the section calculating petrogenic and biospheric end-members for cutbank and point bar samples and removed the calculation of *in situ* biospheric OC. We address the reviewer comments in full in the attached pdf, and thank the reviewers for their time in making detailed and constructive comments on our study.

Synopsis

The primary focus of this study is to constrain the source, age, and flux of carbon that is entrained, transported, and deposited along an Arctic river underlain by discontinuous permafrost (Koyukuk River, Alaska, USA). In particular, the authors collected a suite of samples from eroding cut banks, depositional point bars, and floodplain deposits and compared their grain size, organic carbon content (TOC), and organic carbon $\delta^{13}C$ and $^{14}C$ activity (Fm). They found that finer-grained material is associated with higher TOC and higher Fm, but that carbon stocks are statistically identical in cut banks, point bars, and floodplains. As a consequence of this result, the authors argue that a large fraction of mobilized OC is redeposited and aged during fluvial transport (independent of permafrost cover), rather than being oxidized to $CO_2$.

Overall, I find this study to be highly topical and relevant to an import carbon-cycle question. In particular, I find the combination of techniques taken from organic geochemistry (i.e., $\delta^{13}C$ and Fm) as well as geomorphology/sedimentology (i.e., migration rates, stratigraphic columns, etc.) to be an exciting contribution. That said, there are several statements and calculations that are contradicting, circular, or warrant further clarification—these particularly relate to the interpretation of Fm results. Most importantly: (i) the calculation of individual $Fm_{bio}$ estimates for each sample requires the authors to input $TOC_{petro}$, which is calculated by assuming a constant $Fm_{bio}$ for all samples—this is circular logic. And (ii) the calculations used herein lead to incorrect (artificially high) estimates of the fraction of OC that is produced *in situ*. Still, these issues do not invalidate the main conclusion of the paper, which is that OC stocks in eroding cut banks and deposited point bars are statistically identical, independent of the presence of permafrost.

I detail these points below, along with some minor (line-item) comments. Once these issues have been fully addressed, I support publication of this manuscript in *ESurf*. Please do not hesitate to contact me for further details regarding this review.

Sincerely,

Jordon Hemingway
jordon.hemingway@erdw.ethz.ch

**Reply: Thank you for your detailed review and constructive comments on our manuscript. We agree with your concerns about calculating individual sample $Fm_{bio}$ and have revised our approach and discussion sections to address these concerns. The *in situ* OC calculation requires many assumptions in choosing end-members and does not significantly add to the main story in the manuscript, so we decided to delete this calculation. We describe our changes below, with line numbers referring to the marked-up manuscript file.**

Detailed comments

L14-15: "Radiocarbon content" and "radiocarbon abundance" should be replaced by "radiocarbon activity," since this is the unit of currency used in radiocarbon measurements.
**Reply: We changed to "radiocarbon activity" in L15-16 and throughout the manuscript.**

L14-15: "TOC" should be "TOC content" or "TOC abundance" or similar.
**Reply: We changed to "OC content" (L14) and "TOC content" (L16), as well as throughout the manuscript (for instance, in the Figure 4 caption).**

L75: This should read, "…fluxes of OC into *and out of* the river…"
**Reply: We made this clarification.**

L85: This is not the correct definition of Fm. This should instead be:

$$Fm = \frac{A_{s,N}}{0.95 \times A_{Ox,N}},$$

where $A_s$ is the $^{14}C$ *activity* of the sample, $A_{Ox}$ is the $^{14}C$ *activity* of the Ox-I oxalic acid standard, "N" indicates that both activities are normalized for isotope fractionation (to $\delta^{13}C$ = -25‰ for the sample and -19‰ for Ox-I), and the scalar 0.95 is added by convention. See Reimer et al. (2004) *Radiocarbon* **46**, 1299–1304. Additionally, it sounds a bit strange to say "…with low fractions *of* modern radiocarbon…" I suggest rewording to "…with OC containing low radiocarbon activity, expressed as fraction modern (Fm = …"
**Reply: Thank you for catching our mistake. We corrected the radiocarbon fraction modern definition and implemented your suggested change in wording on L145-148, referencing Reimer et al (2004).**

L109: "…with a constant mass fraction of petrogenic OC." State why $OC_{petro}$ weight percent should be constant—i.e., since it is all eroded from the same bedrock units, independent of grain size.
**Reply: The reviewer makes a good point that our previous justification for a constant $TOC_{petro}$ was insufficient. We think this was in part because the assumptions and equations were introduced out-of-order in the manuscript. In re-doing our mixing model calculations, we relaxed the assumption that all samples must have the same $TOC_{petro}$ and instead report the 95% confidence interval range of $TOC_{petro}$ fitted separately to cutbanks and pointbars (Fig. 4a). This requires the less strict assumption that all cutbanks have the same $TOC_{petro}$ end-member and all point bars have the same $TOC_{petro}$ end-member. We then discuss these assumptions in L214-224.**

L111-120: This set of equations (and possibly the subsequent regression and results) is not strictly accurate. If I read this right, Eq. (4) simply reduces to "$Fm_{meas} = Fm_{bio}(1 - f_{petro})$". The authors state that the mass fraction of $OC_{petro}$ is constant (which is reasonable), but this is not the same as $f_{petro}$ being constant! Rather, $f_{petro}$ is the fraction *of total* OC that is petrogenic. This should instead be rewritten as

$$TOC_{meas} = TOC_{bio} + TOC_{petro},$$

and

$$TOC_{meas} \times Fm_{meas} = TOC_{bio} \times Fm_{bio} + TOC_{petro} \times Fm_{petro},$$

such that

$$Fm_{meas} = Fm_{bio} - TOC_{petro} \times Fm_{bio} \left(\frac{1}{TOC_{meas}}\right),$$

Then, a plot of $TOC_{meas}$ (or $1/TOC_{meas}$) vs. $Fm_{meas}$ yields an intercept that is $Fm_{bio}$ and a slope (or curvature) that is proportional to $TOC_{petro}$, which is the metric that the authors assume to be constant across all samples.
**Reply: The reviewer is correct that we assumed constant $TOC_{petro}$, and we appreciate that our presentation Eq. (1)-(4) is misleading and implies we assumed constant $f_{petro}$. To address these concerns, we revised Eq. (2)-(4) to the form recommended by the reviewer and corrected L213-215 to say that we calculated "$Fm_{bio}$ (…) and the $TOC_{petro}$ content in cutbank and point bar sediment samples."**

L121: Right, here it is clear that TOC$_{petro}$, not *f$_{petro}$*, is constant!
**Reply: Thank you for pointing out this inconsistency in our description of our mixing model. We edited the previous sentence (described above) to correct this mistake.**

L130: Note that this mixing model is only valid if Fm$_{bio,is}$ > Fm$_{bio}$, which might not always be the case due to the decreasing $^{14}$C activity of the atmosphere since nuclear bomb testing ended in the 1960s. That is, biospheric OC fixed ~60 years ago will have a higher Fm value that *in situ* biospheric OC being fixed today. Incorporation of OC fixed over the past ~60 years would therefore artificially increase the calculated fbio,is. These complexities need to be addressed and discussed.
**Reply: We appreciate the reviewer's concerns about the mixing model between *in situ* and re-deposited biospheric OC. Since this calculation does not significantly contribute to our key results and has high uncertainties due to assumptions made when calculating Fm$_{bio}$ and the choice of end-members for cutbank and *in situ* OC, we decided to cut it from the manuscript.**

L131: "…Fm$_{bio}$ is the fraction modern of biospheric OC for each sediment sample…" But won't this be the same for all samples, since a regression of Eq. (4) yields only a single Fmbio value for the whole sample set? Then, if Fm$_{bio,is}$ and Fm$_{bio,cb}$ are assumed to be constant, I don't see how Eq. (5) doesn't yield a constant value for *f$_{bio,is}$* for all samples.
**Reply: The reviewer is correct that using the mixing model assumed a constant Fm$_{bio}$ so that using the TOC$_{petro}$ from the mixing model to calculate individual sample Fm$_{bio}$ is circular. To address this point, we instead do regressions and calculate a range of end-members for cutbank and point bar samples separately. We find that cutbank and point bar samples have similar best-fit TOC$_{petro}$ of 0.100 and 0.075 wt%, and that cutbanks have a slightly younger biospheric end-member (Fm$_{bio}$ = 0.837) than point bars (Fm$_{bio}$ = 0.742). The 95% confidence intervals of these values overlap significantly, as shown in Fig. 4c. In addition, superimposing the range of woody debris Fm values indicates that there is a wide range of plausible biospheric OC end-members, so calculating a single one using the regression may not be representative. Therefore, we compare the 95% confidence intervals for cutbank and floodplain end-members to evaluate the range of Fm$_{bio}$ and TOC$_{petro}$ values. We find that cutbanks and floodplains span a similar range of Fm$_{bio}$ end-members, and that aged Fm values are present in newly deposited point bars. Though this new approach does not allow a quantitative comparison of cutbank and point bar radiocarbon activities, it acknowledges the scatter in the biospheric OC radiocarbon activities and is in agreement with our previous interpretations.**
**Descriptions of this method are found in L212-217 and L353-367; results are presented in L369-374 and Fig. 4c; Fig. 5c was removed and figure panels re-labeled; and the conclusions were updated in L707-709.**

L134: But is the "oldest woody debris from a cutbank" actually representative of the cutbank end member? Should this not instead be the average Fm value of all cutbank samples (or at least the average of the cutbank woody debris samples)? Again, the choice used by the authors here will artificially increase the calculated *f$_{bio,is}$* (i.e., since the true Fm of all OC inputted from the cutbank is almost certainly higher than that of the oldest woody debris). This should be discussed.
**Reply: We acknowledge that the choice of end-members will significantly impact the calculation of how much OC is oxidized during sediment transport. Since the *in situ* OC calculation builds on the uncertainty of calculating Fm$_{bio}$ for each sample and it is not obvious how to select appropriate end-members, we have decided to cut this calculation from the revised manuscript. Therefore, we deleted L205-211 and Supplemental Table S6, revised Supplemental Tables S3 and S5, and re-numbered equations throughout the manuscript. We also added the range of woody debris radiocarbon Fm values to the y axis of Fig. 4c.**

L189: Which values—LANL or NOSAMS—were used in all subsequent calculations? For some metrics in some samples, this appears to make quite a difference.
**Reply: We clarified that all subsequent calculations were done using the LANL measurements in L311-313. We chose to use the LANL measurements because correspondence with technicians at WHOI-NOSAMS indicated that they recommended separately analyzing samples for TOC and using TOC content measurements from their facility for comparative purposes.**

L217: This should read "Sediment TOC and radiocarbon Fm varied…" (remove "measurements")
**Reply: We removed "measurements" in L345.**

L231-232: Can the authors elaborate on what they mean by "Fitting to TOC:TN weight ratios…"? Wouldn't this yield the (somewhat odd) result of $TOC_{petro}:TN_{meas}$? I'm confused by the motivation or benefit of calculating the regression in this way.
**Reply: We apologize for the confusion. You are correct that we tested fitting $Fm_{meas}$ vs $TOC_{meas}:TN$ weight ratios to calculate $Fm_{bio}$ assuming constant $TOC_{petro}:TN_{petro}$, but found a poorer correlation and therefore did not use those fitted end-members. Since this result is not used in subsequent analyses and does not appear in any figures, we deleted this sentence (L363-364) for clarity.**

L249-251: But the regression in Fig. 4d requires the assumption of a constant $Fm_{bio}$ across all samples in order to calculate $TOC_{petro}$. So, how can this result then be used to estimate variable $Fm_{bio}$ values for each sediment sample? This seems circular and will, by definition, yield estimated $Fm_{bio}$ values that are the same across all grain size classes.
**Reply: We agree with the reviewer that these assumptions are circular. We updated the manuscript to present a range of plausible end-members for cutbank and point bar sediment samples in L369-374 and Fig. 4c, as described above.**

L273: Remove one of the redundant "of" instances
**Reply: We deleted the repeated "of."**

Fig. 4: Again, I'm not really sure $Fm_{bio}$ can be calculated for each individual sample, when the calculated inputs for this calculation (i.e., $TOC_{petro}$) require the assumption of constant $Fm_{bio}$ across all samples (i.e., Eq. 4). This seems circular to me.
**Reply: The reviewer is correct, and we have eliminated calculating *Fm~bio~* for individual samples, instead calculating a range of end-members for samples grouped cutbanks and point bars separately, and updated Fig. 4 with the new results.**

L349-350: Clarify by changing to "…reflects the greater *proportional* petrogenic OC contribution in coarser material"
**Reply: Thank you for the suggestion – we added "proportional" in L555 for clarity.**

L350-352: How can biospheric OC simultaneously have a "similar Fm for all grain sizes" but "fine sediment … contain higher $Fm_{bio}$"? These statements are contradicting. I'm also not convinced by the evidence for *in situ* biospheric production (as detailed above and below).
**Reply: We agree with the reviewer that these sentences are confusing, and revised this section to discuss results from the new *TOC~petro~* and *Fm~bio~* regressions for cutbank and point bar samples (L553-558).**

L354: Space between "being present"
**Reply: We corrected the typo.**

L376-380: As mentioned in my above comments (L130, L134), this calculation will lead to an artificially high estimate of $f_{bio,is}$. Rather, the average $Fm_{bio}$ for cut bank inputs—which is more representative for the calculation of $f_{bio,is}$—is much higher than $0.2319\pm0.00152$.

This again comes back to the regression from Eq. (4) used to calculate a constant $Fm_{bio}$ and $TOC_{petro}$ across the entire sample set. Given this calculation, all samples will by definition have the same calculated $Fm_{bio}$ and thus there is no "room" to oxidize and replace some $TOC_{bio}$ with $^{14}C$-enriched *in situ* material.

Furthermore, how is it possible that "cut banks have 72% of sediment TOC produced *in situ*" (L380), but cut bank $Fm_{bio}$ ($Fm_{bio,cb}$) is also used as the non-*in situ* end-member in Eq. (5) (L130)? According to Eq. 5, shouldn't cut banks have 0% *in situ* TOC by definition? This needs to be clarified and discussed further.

I am also now somewhat confused as to what specifically the authors mean by "*in situ*" production. Does this refer only to TOC produced during transport within the river itself (I think this is the case), or does this refer to any non-permafrost $TOC_{bio}$ (which I think is what is implied by using the oldest woody debris for the non-in situ end member Fm value)? If the former, then why would the authors expect their results here to deviate so far from previous estimates (e.g., Scheingross et al., 2021), which conclude that there is very limited oxidation and *in situ* production during fluvial transport?

I think the assumptions and calculations used to calculate $f_{bio,is}$ need to be critically reevaluated. This also warrants updating some discussion points later in the manuscript (e.g., L390, L398, L461)

**Reply: We have removed the calculation of $f_{bio,is}$ from the manuscript due to the concerns about the mixing model raised by both reviewers. To address these specific points of concern, we changed L709 from "approximately one-quarter to one-half" to read "a portion of". We significantly revised L577-595 to discuss biomass production on point bars qualitatively, instead of using the mixing model. In L602-603 we deleted "based on the estimated contribution of *in situ* production on point bars". We also corrected the equation numbering throughout the manuscript and deleted Supplemental Table S6.**

Summary

This manuscript explores the effect of riverbank erosion on the fate of permafrost organic carbon, specifically examining the balance between removal of OC from permafrost cutbanks and deposition of OC onto point bars in the Koyukuk River, Alaska. The key question being asked is whether OC mobilized from ancient permafrost deposits is oxidized to CO2 within river systems or is it quickly re-deposited and buried in floodplains. This study is timely because the effects of climate change on the arctic carbon sink cannot yet be predicted, and studies like this will enable us to predict changes in permafrost carbon stocks as river channel migration rates accelerate in the future. This is a relevant scientific question within the scope of *ESurf*.

The authors quantified OC stocks and used a 1-D mass-balance model to quantify net fluxes of OC from cutbanks, to point bars, and to downstream transport. The estimation of carbon stocks and the balance of OC fluxes in the river system appear robust, however, the estimation of OC produced by floodplain vegetation needs to be revisited. The authors should also revisit the biospheric fraction modern values calculated for individual samples. These issues do not appear to affect the overall message of the manuscript, which is that biospheric OC production on point bars is sufficient to offset any OC lost to downstream transport or OC oxidation in the floodplain. However, there are flaws in their calculations that need to be corrected.

Overall, I think this is a nice study that will be of interest to readers of *ESurf*. However, details regarding some measurements and calculations need to be revised before this manuscript can be accepted. Therefore, I support publication after the authors address the concerns detailed below.

**Reply: Thank you for your helpful review of our manuscript. We revised our approach for calculating biospheric fraction modern values and biospheric OC production on point bars in response to your concerns. We describe these revisions and others in more detail below, with line numbers referring to the marked-up manuscript.**

Detailed comments

L15: change *radiocarbon abundance* to *radiocarbon activity*
**Reply: We changed "radiocarbon abundance" to "radiocarbon activity" in L15-16 and throughout the manuscript.**

What is the timescale over which sediment is deposited onto the point bar?
**Reply: An upper bound on the timescale of bar deposition is the time for the Koyukuk River to migrate one channel width. Using the migration rate and a channel width of 175 m (Figure S2), the timescale for river migration is T ~ channel width / river migration rate ~ 175 / 0.52 m yr$^{-1}$ ~ 340 yr. Alternatively, a more appropriate timescale may be the time it takes for the river to deposit a package of sediment at a location on the point bar equal to its depth, since much of the channel bed is continually re-worked by the river. Using the channel depth and migration rate, a timescale for local sediment deposition is T ~ channel depth / river migration rate ~ 12.4 m / 0.52 m yr$^{-1}$ ~ 24 yr. Once that sediment has been deposited, it may remain in place without being re-mobilized by bank**

**erosion for timescales up to ~10 kyr, 2-3 orders of magnitude greater than our estimates of the deposition timescale.**

What is the role of petrogenic OC? Is it possible that petrogenic OC gets preferentially deposited on point bars because it is associated with the denser mineral fraction, while aged biospheric organic matter is transported downstream due to its lower density?

**Reply: We agree with the reviewer that sediment with high proportions of petrogenic OC versus aged biospheric OC from permafrost likely has different physical properties (density, grain size, propensity to flocculate, etc.) that may affect its transport and storage in river systems. Previous work has shown that $OC_{petro}$ content may vary with sediment properties that depend on grain size. The $OC_{petro}$ content of eroding black shales depends on mineral surface area, clay chemistry, and $OC_{petro}$ being sandwiched between layers of clays (Kennedy et al., 2002). Clay abundance, surface area, and cation exchange capacity vary with grain size, and resulting differences in $OC_{petro}$ loading can persist and have cascading effects far downstream (Blattmann et al., 2019). For instance, $OC_{petro}$ (wt%) decreases with increasing Al/Si ratio for suspended sediments in the Yangtze River, and most $OC_{petro}$ being associated with coarse grains leads to hydrodynamic sorting in marine settings (Sun et al., 2021). However, unlike in marine deposits, rivers re-visit and entrain previously deposited sediment. Therefore, preferential transport of $OC_{petro}$ in coarse-grained bedload and deposition in bars would increase the number of sediment storage events in the floodplain, thus increasing the sediment transit time, but they should not cause a net sink of $OC_{petro}$ into the river floodplain without long-term aggradation of the river reach.**

**In our study, we primarily focus on petrogenic OC as an explanation for aged radiocarbon signatures in river networks independent of the presence of permafrost in the watershed. However, petrogenic OC plays a distinctive role from biospheric OC in the global carbon cycle. In general, oxidation of $OC_{petro}$ is a long-term source to the atmosphere while its reburial in river deltas generates no net effect on atmospheric $CO_2$. Biospheric OC instead can provide a long-term $CO_2$ sink if it is buried in aggrading sedimentary systems such as river deltas (e.g., Galy et al., 2007).**

**We infer that the reviewer is concerned that the differing physical properties between biospheric and petrogenic OC may cause point bar sediment to be enriched in $OC_{petro}$ relative to cutbank sediment. We do not anticipate that the cutbanks have different $OC_{petro}$ content than point bars, because the cutbanks are comprised of old point bar deposits that were abandoned by the river and are now being re-visited and eroded. To address this concern, we also allowed the cutbanks and point bars to have different *$TOC_{petro}$* content end-members in our revised mixing model (L212-217). The 95% confidence intervals of plausible cutbank and point bar petrogenic OC end-members overlap, supporting our hypothesis that floodplain sediment has similar petrogenic OC content.**

**References:**
**Blattmann, T. M., Liu, Z., Zhang, Y., Zhao, Y., Haghipour, N., Montluçon, D. B., Plötze, M., and Eglinton, T. I.: Mineralogical control on the fate of continentally derived organic matter in the ocean, https://doi.org/10.1126/science.aax5345, 2019.**
**Galy, V., France-Lanord, C., Beyssac, O., Faure, P., Kudrass, H., and Palhol, F.: Efficient organic carbon burial in the Bengal fan sustained by the Himalayan erosional system, 450, 407–410, https://doi.org/10.1038/nature06273, 2007. Kennedy, M. J., Pevear, D. R., and Hill, R. J.: Mineral Surface Control of Organic Carbon in Black Shale, Science, 295, 657–660, https://doi.org/10.1126/science.1066611, 2002.**
**Sun, X., Fan, D., Cheng, P., Hu, L., Sun, X., Guo, Z., and Yang, Z.: Source, transport and fate of terrestrial organic carbon from Yangtze River during a large flood event: Insights from**

multiple-isotopes (δ13C, δ15N, Δ14C) and geochemical tracers, Geochimica et Cosmochimica Acta, 308, 217–236, https://doi.org/10.1016/j.gca.2021.06.004, 2021.

L85: This equation for fraction modern is incorrect. See Jordan Hemingway's review for reporting the correct equation. Equation 1 in Reimer et al. (2004) is the full equation. Other useful references for fraction modern reporting are: Stuiver and Polach, 1977; Donahue et al., 1990. I also recommend rephrasing "low fractions of modern radiocarbon" to "low radiocarbon activity"

**Reply: Thank you for correcting our error in the fraction modern equation. We corrected the equation and added the reference to Reimer et al. (2004) in L146-148.**

L87: "…low Fm *values* inherited…"

**Reply: We added "values" in L159.**

L91: Avoid saying "low or high Fm carbon." Alternatively, I would recommend saying "organic carbon with low Fm values", or "radiocarbon depleted OC".

**Reply: Thank you for the suggestion for improved wording. We changed L153 from "low Fm carbon" to "organic carbon with low Fm values".**

L104: TOC "concentrations" and Fm "values." Fm is not a measurement, but a value calculated from measurements of 14C and 12C.

**Reply: We made the suggested change to "Fm values" in L176, and changed to "TOC content" (since that specifically refers to mass of a solid) throughout the manuscript.**
**Reference: IUPAC: Compendium of Chemical Terminology, 2nd ed. (the "Gold Book"). Compiled by A. D. McNaught and A. Wilkinson. Blackwell Scientific Publications, Oxford (1997). Online version (2019-) created by S. J. Chalk. ISBN 0-9678550-9-8. https://doi.org/10.1351/goldbook.**

L107: In the parentheses, do you mean to say $OC_{petro}$ and $OC_{bio}$, rather than $TOC_{petro}$ and $TOC_{bio}$?

**Reply: Thank you for pointing out this confusion in L180-181. Throughout the manuscript we tried to have acronyms in italycs reflect variables used in our calculations, so these were intended to refer to $TOC_{petro} = f_{petro}TOC_{meas}$ and $TOC_{bio} = f_{bio}TOC_{meas}$. However, this was not clear in the originally submitted manuscript.**

L108: TOC "concentrations"

**Reply: We added "contents" in L180 and throughout the manuscript.**

L110: Figure 4d does not exist.

**Reply: Thank you for catching this. L181 should refer to Figure 4c but was incorrect from an earlier draft.**

L110: Constant $f_{petro}$ or $TOC_{petro}$? Make sure the appropriate terms are used throughout the text.

**Reply: Thank you for indicating we were not clear in describing our methods. In L194-195 we added parentheses to clarify that we assumed constant $TOC_{petro}$ and allowed variable $f_{petro}$. Reviewer 1 (Jordon Hemingway) also raised concerns about the clarity of this section, and we describe additional edits to L178-223 in our response to his comments.**

L127: Briefly define what you mean by *in situ*.

**Reply: We appreciate that "*in situ* biospheric OC" was not adequately defined in our submitted manuscript. We had intended to separate out sediment OC that was deposited on a point bar from OC fixed by primary production on the point bar after it was built. Therefore, *in situ* biospheric OC could include both permafrost OC (such as moss that grew and was subsequently frozen for many years) and non-permafrost OC in old river deposits. Since assigning end-members for OC that survived transport versus newly fixed biospheric OC is complex and the biospheric $^{14}$C end-member will vary significantly through the bomb spike, we decided to cut this section of analyses.**

L134: Why choose the oldest woody debris as the end-member? The permafrost OC is a mixture of organic matter varying in 14C activity, and as such should be represented by a mean value for ancient permafrost OC samples/woody debris samples. By using the oldest-value as the end-member, the mixing model will result in overestimated proportions of $OC_{bio,is}$.
**Reply: We agree with the reviewer that assigning an end-member for cutbank versus *in situ* biospheric production is not clear and likely overestimated $OC_{bio,is}$, so this section was cut.**

L157: 12.4 m? Need units here.
**Reply: Added appropriate units (meters) in L277.**

Section 3.1 Field sampling methods – it's not clear why the paragraphs are broken up like this. I would combine the ADCP depth with the channel migration rate calculations and move the digging/coring sampling method immediately after the categorization of permafrost vs. non-permafrost.
**Reply: Thank you for the suggestion. We re-arranged the text in Section 3.1 accordingly to better integrate ideas and methods.**

L167: SIPRE "corer"
**Reply: Corrected to "SIPRE auger" (https://icedrill.org/equipment/hand-auger-sipre).**

L170-180: Were the samples not ground to a fine powder prior to EA and EA-IRMS analyses? I worry that the samples were not thoroughly homogenized prior to analysis. This is particularly important for permafrost cutbank samples that contain both mineral-associated OC and particulate plant debris.
**Reply: We apologize for the reviewer's confusion on our sample processing. The samples were not ground to a fine powder before their initial split because we needed to measure grain size on one split, so we made those splits to avoid fractionating by grain size. Then, we ground the splits for geochemical analyses again using a mortar and pestle to properly homogenize them. We clarified this procedure in L286, L290, and L303.**

Additionally, TOC and TN concentrations would be measured on the elemental analyzer, and stable carbon and nitrogen isotopes would be measured on the EA coupled to the IRMS. Please clarify that both concentrations and isotopic compositions were measured and on which instruments.
**Reply: Thank you for pointing out this omission. The measurements were all conducted simultaneously on the EA coupled to the IRMS. We revised L288-289 to say, "Total organic carbon *content* (…), *stable organic carbon isotopes,* and total nitrogen (TN) *content* were measured…".**

I also suggest moving the details about the measurement calibration standards to the supplement, but it is also fine if you leave it in.
**Reply: We appreciate that the details about calibration standards may not be of interest to all readers of the manuscript. However, since the supplement currently contains data tables and no**

**other methodological information, we prefer to leave all of the methods together in the main text so they can be easily located by the reader.**

L187: It seems confusing that you had TOC concentrations and stable isotope values measured at two different labs. Please add a sentence before L170 that briefly describes why samples were acid-treated using two different methods and then measured at two different facilities. Were the duplicate EA and EA-IRMS analyses performed on all samples? Additionally, please explain why you used the LANL TOC concentrations for the $TOC_{meas}$ values.

**Reply: The TOC concentrations and stable isotope values were measured at both LANL and NOSAMS because we were interested in measuring TN as well as TOC (conducted at LANL) and NOSAMS does not formally report TOC values. NOSAMS measured TOC as reference for determining the sample amount required for $^{14}C$ measurement. We had requested TOC values from NOSAMS from the technicians because we were concerned about comparing OC measurements between labs with different decarbonation methods, and they responded with values but noted a very high measurement uncertainty in TOC values (5 wt%, L209). Due to its expense, only a subset of samples were analyzed for radiocarbon activity, while more samples were analyzed at LANL for TOC, $\delta^{13}C$, and TN. To make our analyses more statistically robust, we chose to use the LANL measurements for TOC concentration in our OC budget, and to discuss potential issues with incomplete decarbonation of the NOSAMS samples when analyzing our radiocarbon data (L310-311). In summary, we find that including these low Fm, high $\delta^{13}C$ samples may cause us to under-estimate the extent of oxidation of young, labile OC, but does not change our key findings. To address the reviewer's concerns, we made clear in L311-313 which analyses were used in each calculation.**

L196: What was the percent concentration of Calgon solution?

**Reply: The Calgon solution was 10 g sodium hexametaphosphate per L DI water, now described in L321.**

L197: Were samples split twice? L191 says samples were split using a riffle splitter.

**Reply: We appreciate the reviewer's confusion in tracking sample splits. Our dried samples consisted of up to ~200g of dried sediment, which we split into 4-8 splits (depending on the sample size) using a riffle splitter or via cone and quartering (described in L286). We needed to generate multiple splits of the samples to send some to LANL for carbon and nitrogen analyses. One of those splits was then subsampled to the size required for pre-treatment for grain size analysis (L315), again using a riffle splitter. After pre-treatment, the grain size samples were split again using a riffle splitter while wet to an appropriate sediment concentration for measurement (L322-323). This additional wet split was required because the Malvern Mastersizer can only make measurements in a range of obscuration values, and it is very hard to estimate an appropriate amount for the measurement prior to pre-treatment.**

**To clarify how we made splits, we changed L315 to read "Sample splits for grain size analysis were placed…", since the riffle splitting and cone and quarter methods are already described in L282. The wet splitting is clarified in L322-323: "The samples were split while wet using a riffle splitter to the required sediment concentration for laser diffraction, and grain size was measured on a Malvern Mastersizer 2000, with measurements…"**

L199-200: Is it necessary to say that sand card grain size approximations were made? If you only use the laser diffraction particle size data, then the non-quantitative field observations seem irrelevant.

**Reply: We apologize for the reviewer's confusion on this point. We used field observations of grain size to obtain grain size measurements at small increments (1 cm resolution) across all of our stratigraphic columns (~16 m of core, 5 m of soil pits, and 30 m of cutbank). It was not feasible to sample or measure grain size with this density, so we used laboratory measurements of grain size to confirm that our field observations of grain size were accurate. This allowed us to group our OC concentrations by measured grain size class and extend these measurements to meters of core across the landscape.**

**To make this intention more clear, we revised L325-326 to read "Grain size data confirmed our field observations of grain size that were made using a sand card and hand lens,."**

L217: TOC "content" or "concentrations"
**Reply: We changed "TOC" to "TOC content" where the reviewer indicates and throughout the rest of this paragraph (L345-349).**

Figure 4: Using the term "organics" is somewhat misleading on the plots. I suggest changing to "organic horizon." It would also be helpful if the shaded regions were labeled with blue text, red text, etc. for the material they represent.
**Reply: We changed "Organics" to "Organic horizon" and labeled the shaded regions with their stratigraphic unit on Fig. 4.**

L222: yielded lower "Fm values"
**Reply: Changed to "Fm values" in L352.**

L225: to calculate the "proportions" or the "radiocarbon activity" of petrogenic and biospheric end-members?
**Reply: Thank you for the clarifying suggestions. We were assuming that the petrogenic end-member has zero radiocarbon activity, and intended to calculate the proportions of this petrogenic and a biospheric end-member. We changed L354 to read "To calculate _the range of TOC_$_{petro}$ _and Fm_$_{bio}$ end-members, ..." in accordance with the reviewer's recommendation.**

Why must there only be two end-members? From my perspective, there is a petrogenic end-member, a permafrost-derived aged biospheric end-member, and a modern biospheric end-member. Because you sampled permafrost OC from cutbanks and soil organic horizon OC, you should be able to sufficiently characterize these three end-members.
**Reply: The reviewer is correct that there is a range of Fm values for the biospheric end-members. In the initial submission, we were not clear in defining and justifying our choice of end-members, and therefore substantially revised this section in response to comments from reviewer 1. After sediment is deposited in a point bar, it may re-form permafrost, which generates permafrost OC of varying age across the floodplain. Since river deposits have varying age, shown by cross-cutting relations between scroll bars, we would not expect a single age of permafrost formation. This prevents us from using a given sediment sample as permafrost-derived aged biospheric end-member, and the wide range of deposit ages is reflected in the wide range of Fm values for woody debris taken from sampling locations with and without permafrost.**

**To address this limitation and make our _Fm_$_{bio}$ calculations analogous to our flux calculations, we instead calculated petrogenic and biospheric end-members for the cutbank and point bar sediment separately (L369-374) and describe how permafrost may contain OC stocks of**

**varying ages depending on when sediment comprising that area of the floodplain was deposited (L154-156). We also revised Fig. 4c and added the range of woody debris Fm values on the y axis.**

L229-230: Supplemental figure S4 contains field photos. I think you want to reference Fig. S5. I'm confused about the exclusion of >-20 per mil because all the d13C values on those plots are < -23 per mil – did you exclude the elevated d13C samples from Figures S5 and S6 as well? And it looks like the samples with higher TOC content do not have elevated d13C values, so I don't think incomplete carbonate removal is a concern. However, it is interesting that the NOSAMS d13C values were often significantly higher than the LANL d13C values. I worry that because the radiocarbon measurements were made on the same aliquots as these elevated d13C measurements, then the radiocarbon activities would also be influenced by any remaining carbonate.

**Reply: We share the reviewer's concerns about the potential for incomplete decarbonation affecting the radiocarbon measurements, and also that our presentation in this section is unclear. To assess whether incomplete decarbonation would bias our fitted *Fm$_{bio}$* and *TOC$_{petro}$* values for cutbanks, we combined cutbank and floodplain samples and excluded cutbank samples with $\delta^{13}C >$ -20 per mille and found little effect on our results (L359-367). To clarify our reasoning we also revised the caption for Fig. S5 and S6 and changed the parenthetical reference to Table S2.**

L229: Fig. 4c
**Reply: Changed to Fig. 4c.**

L252: If f$_{petro}$ varies across samples, but you are still using the regression to calculate Fm$_{bio}$ for individual samples, you will end up calculating similar Fm$_{bio}$ values for all samples. You would need a different end-member mixing model approach to calculate f$_{bio}$ for individual samples.
**Reply: We agree with the reviewer, and have instead fit a range of representative *Fm$_{bio}$* end-members for all cutbank and point bar samples, with results described in L369-374.**

L280: It might be helpful to write out the assumption that M$_{H2O}$ + M$_{dry}$ = 1.
**Reply: We agree that it was not clear that M$_{H2O}$ indicates a dimensionless mass fraction of water in sediment. We write out this assumption explicitly in L467-468.**

L340: How do these OC fluxes compare with the OC flux exported by the river? A more thorough mass-balance would account for the sediment flux in the river and determine how much is deposited onto point bars over the ~80m reach studied. Although, it may not be necessary if the main point is that modern biospheric production and soil development on point bars is the primary mechanism of balancing OC stocks between cutbanks and point bars.
**Reply: We agree with the reviewer that directly comparing bank erosion and downstream OC fluxes would strengthen our discussion. Unfortunately, there is no previously published particulate OC flux for the Koyukuk catchment, and the hydrology is poorly constrained because there is not currently a USGS gage in operation on the Koyukuk main steam.**

 **For a back-of-the-envelope comparison, the Koyukuk River at Hughes had an annual suspended sediment load of $2 \times 10^6$ tons ($1.81 \times 10^6$ metric tons) during the intervals in the 1980s when the USGS gage was maintained, and the Yukon River at Pilot Station carries $60 \times 10^6$ tons ($54.4 \times 10^6$ metric tons; Brabets et al., 2000). The Yukon River exports $(218 \pm 24) \times 10^9$ g of particulate OC per year, measured using a discharge-weighted average at Pilot Station (McClelland et al., 2016). Assuming that the concentration of suspended sediment is proportional to the concentration of particulate OC (POC) and that the suspended sediment load of the Koyukuk River**

has been constant since the 1980s, we expect the Koyukuk to export $(7.3\pm0.8)\times10^6$ kg/yr of POC. Differencing cutbank versus point bar OC stocks gives a minimum downstream flux of 5.9 kgOC/yr/m downstream, which requires 1200 km of meandering river length to generate the downstream POC flux for the Koyukuk River. If instead we use our upper estimate (Discussion) of 30 kgOC/yr/m downstream POC flux, we find a meandering river length of 240 km is sufficient to generate the POC flux of the Koyukuk. For comparison, the linear distance between the river headwaters at Wiseman, AK and confluence with the Yukon River at Koyukuk, AK is approximately 440 km. These back-of-the-envelope calculations indicate that our calculated fluxes for bank erosion are reasonable in comparison to the estimated downstream POC flux for the Koyukuk, but we chose not to include this calculation in our discussion because the downstream POC flux for the Koyukuk River is very poorly constrained.

To address this knowledge gap and constrain the net OC flux downstream in our study reach, we collected dissolved OC, suspended sediment, and bed sediment samples and river hydraulic measurements. While those samples are still being analyzed, a preliminary comparison of cutbank versus suspended sediment OC contents indicates a partial loss of particulate OC to oxidation but preservation of aged OC through transport, in agreement with our findings in this manuscript.

**References:**
Brabets, T. P., Wang, B., and Meade, R. H.: Environmental and hydrologic overview of the Yukon River Basin, Alaska and Canada, US Department of the Interior, US Geological Survey, 4204 pp., 2000.
McClelland, J. W., Holmes, R. M., Peterson, B. J., Raymond, P. A., Striegl, R. G., Zhulidov, A. V., Zimov, S. A., Zimov, N., Tank, S. E., Spencer, R. G. M., Staples, R., Gurtovaya, T. Y., and Griffin, C. G.: Particulate organic carbon and nitrogen export from major Arctic rivers, 30, 629–643, https://doi.org/10.1002/2015GB005351, 2016.

L354: need space between *being* and *present*
Reply: We corrected this typo.

L380: I think the authors need to re-calculate the percentages of *in situ* biospheric OC in the samples. It seems highly unlikely that cutbanks have ~72% of sediment OC produced *in situ* given what we know about permafrost carbon stocks. I think this can be corrected by using a permafrost OC cutbank end-member with a higher mean Fm value.
Reply: In response to concerns raised by both reviewers about calculation the fraction of *in situ* biospheric OC in each sample we have cut this calculation from the manuscript.

L382: Some oxidation of modern, labile OC? Or rather, oxidation of labile ancient permafrost-derived OC?
Reply: We clarify in L589-595 that the radiocarbon data indicates that modern, labile OC is being preferentially oxidized, while OC with lower radiocarbon activity appears to be more recalcitrant.

L414: "biospheric OC *production*?"
Reply: The reviewer is correct that we intended to include primary production in plants and soil microbial communities. We followed their suggestion and changed "input" to "production" in L634.

L455: TOC content and Fm values
**Reply: We corrected the typo.**

L459:
**Reply: Removed "that" from L636.**

L460: The authors should reconsider the calculations of $Fm_{bio}$ for individual samples and not make a comparison across grain sizes using those estimated values.
**Reply: We have redone these calculations and instead calculated bulk end-members for *$Fm_{bio}$* of cutbanks and point bars. We revised L707-709 to read: "Our results indicate that floodplain processes generated an aged biospheric radiocarbon signature in newly deposited point bars, and variations in sediment Fm with grain size may be due to mixing with a petrogenic end-member."**

**Figures**

Figure 1: Is it true that channel width has been remaining constant over the last few decades? Based on Figure S2, it appears that there is a minor shift to increased channel widths, and I wonder if mean values would be better suited to characterize this measurement. Mean values would only be appropriate if the channel width is measured continuously along the study reach.
**Reply: We agree with the reviewer's concern that channel width may not remain constant over decadal timescales. The width is measured continuously along the centerline of the channel reach in Rowland et al. (2019), and that data is plotted in the histogram in Figure S2. The histogram for 1978 includes an upstream reach that is narrow and becomes cutoff from the main channel before 2018, leading to a decrease in channel length. Therefore, the PDF for 1978 gives slightly narrower channel widths than in 2018 that are compensated by a decrease in reach sinuosity to give a constant channel area. This is accounted for in our calculation of average migration rate (L525-531). To clarify this point, we added a discussion point on the effects of transience in river OC fluxes in L683-687.**

Figure 4c: Should the blue squares be blue circles?
**Reply: Thank you for pointing out that the color- and shape-coding for Fig. 4c is incorrect, wrongly implying that there is abundant permafrost on point bars. We re-generated the figure with the correct symbols.**

Figure S1: I recommend using different colors or shapes to distinguish between core, bank, and pit samples, like that used in Figure 2.
**Reply: Thank you for the recommendation. We updated Fig. S1 with the same color- and shape-coding as Fig. 2.**

**References Cited**

Donahue, D. J., Linick, T. W., & Jull, A. T. (1990). Isotope-ratio and background corrections for accelerator mass spectrometry radiocarbon measurements. Radiocarbon, 32(2), 135-142.

Reimer, P. J., Brown, T. A., & Reimer, R. W. (2004). Discussion: reporting and calibration of post-bomb 14C data. Radiocarbon, 46(3), 1299-1304.

Stuiver, M., & Polach, H. A. (1977). Discussion reporting of 14C data. Radiocarbon, 19(3), 355-363.

---

## Author Response (AR2)

Thank you for accepting our manuscript to Esurf. In response to the editor's technical corrections, we have made the following changes, with all line numbers corresponding to the final uploaded manuscript.

*Line 180 (tracked changes version) - add a note after Equation 2 to spell out f_bio and f_petro (i.e. "where f_bio and f_petro are the fraction of organic carbon from biospheric and petrogenic sources", or similar)*

We added the editor's suggested definition for $f_{bio}$ and $f_{petro}$ in L120 after Eq. 2.

*Line 560 - the sentence implies that woody material in has been specifically sampled and targetted. Please rephrase. Also, could this be plant debris, not necessarily wood? (also see elsewhere where the term wood is used)."*

We realize that the previous phrasing implied that we specifically targeted and sampled wood at a broader range of locations than where we sampled floodplain sediment. To address this confusion, we changed L369-370 to read: "This observation matched the range of aged wood and plant debris found at sediment sampling locations." We also changed "woody debris" to "wood and plant debris" throughout the manuscript and in Fig. 4.